# Towards explicit regulating-ion-transport: nanochannels with only function-elements at outer-surface

Qun Ma[1], Yu Li[1], Rongsheng Wang[1], Hongquan Xu[1], Qiujiao Du[2], Pengcheng Gao[1✉] & Fan Xia [1✉]

Function elements (FE) are vital components of nanochannel-systems for artificially regulating ion transport. Conventionally, the FE at inner wall ($FE_{IW}$) of nanochannel−systems are of concern owing to their recognized effect on the compression of ionic passageways. However, their properties are inexplicit or generally presumed from the properties of the FE at outer surface ($FE_{OS}$), which will bring potential errors. Here, we show that the $FE_{OS}$ independently regulate ion transport in a nanochannel−system without $FE_{IW}$. The numerical simulations, assigned the measured parameters of $FE_{OS}$ to the Poisson and Nernst-Planck (PNP) equations, are well fitted with the experiments, indicating the generally explicit regulating-ion-transport accomplished by $FE_{OS}$ without $FE_{IW}$. Meanwhile, the $FE_{OS}$ fulfill the key features of the pervious nanochannel systems on regulating-ion-transport in osmotic energy conversion devices and biosensors, and show advantages to (1) promote power density through concentrating FE at outer surface, bringing increase of ionic selectivity but no obvious change in internal resistance; (2) accommodate probes or targets with size beyond the diameter of nanochannels. Nanochannel-systems with only $FE_{OS}$ of explicit properties provide a quantitative platform for studying substrate transport phenomena through nanoconfined space, including nanopores, nanochannels, nanopipettes, porous membranes and two-dimensional channels.

[1] State Key Laboratory of Biogeology and Environmental Geology, Engineering Research Center of Nano-Geomaterials of Ministry of Education, Faculty of Materials Science and Chemistry, China University of Geosciences, Wuhan, P. R. China. [2] School of Mathematics and Physics, China University of Geosciences, Wuhan, P. R. China. ✉email: pcgao@cug.edu.cn; xiafan@cug.edu.cn

Nanochannel-systems are artificial passages of ions and molecules with unique controllable performances[1,2]. They have been widely used in sensing[3–5], drug release[6,7], separation[8–11], nanofluidic[12], nanoelectrochemistry[13,14], and energy conversion[15–17], owing to their adjustable geometries at the nanoscale, versatile chemical compositions, and strong mechanical strength. The nanochannel-systems usually consist of three components: (1) nanochannels, (2) function elements at outer surface (FE$_{OS}$), and (3) function elements at inner wall (FE$_{IW}$)[18,19]. However, in traditional nanochannel-systems, there are two troublesome "black boxes" which are not well addressed: one is the role of FE$_{OS}$ on ion transport, which has been long-termed neglected; and the other is inexplicit chemical and physical properties of FE located deep inside nanochannel which is subject to that few test techniques with test tips or testing liquids, that can sufficiently contact with FE in the confined space at the nanoscale[2].

Currently, both theoretical[20,21] and experimental investigation[22–25] showed the synergistic effects of FE$_{OS}$ on regulating-ion-transport in the presence of FE$_{IW}$. Compared with the confined space in nanochannel, relatively more free-spaces of OS endow FE$_{OS}$ with advanced characteristics, such as easy to immobilize, available for precise characterizations, receptive for foreign substrates, and potential application in new scenarios. However, till now the properties of most FE are still inexplicit or generally presumed from the measurable FE$_{OS}$[26,27], which would bring potential errors.

Here, we confined FE at the outer surface (OS) and the edge of IW with a minimum depth as 7.5 nm (~0.01% of total IW) in nanochannels through the threshold effect of reducing-diameter down to 11 ± 3 nm, which is detectable for a host of techniques and termed as FE$_{OS}$. The FE$_{OS}$ have been certified to regulate ion transport independently and their mechanism can be well demonstrated through the Poisson and Nernst–Planck equations assigned by measured properties from atomic force microscope (AFM), time of flight secondary ion mass spectrometry (ToF-SIMS), and solid-surface zeta potential analyzer (SSZPA). The FE$_{OS}$ fulfill the key capabilities of nanochannels in osmotic energy conversion and biosensing and bring new features: (1) increase of ionic selectivity but no obvious change in resistance and (2) accommodating probes or targets with size beyond the diameter of nanochannels.

## Results

**Designed nanochannel-systems.** Different from previous nanochannel-systems (Fig. 1a, b), we designed a new nanochannel-system (Fig. 1c). In the 1st stage in Fig. 1a and 2nd stage in Fig. 1b, both the physicochemical properties and function on regulating-ion-transport of nanochannel system are partially unclear due to the limitation for the characterization of FE$_{IW}$. While, using the nanochannel-system only consisting of independent FE$_{OS}$ (Fig. 1c) will avoid addressing the two problems: one is the role of FE$_{OS}$ on regulating-ion-transport and the other is unclear physicochemical properties of FE$_{IW}$. Hence, the explicit relationship between the physicochemical properties of FE and function of nanochannel-systems could be realized.

**Fabrications of nanochannel-system with only FE$_{OS}$.** We built a nanochannel-system using an anodic aluminum oxide (AAO) membrane deposited by Au at the one side as nanochannels (named as none@OS) (Fig. 2a–e, Fig. S1)[22,23]. For the present nanochannels, their surfaces could be divided into two parts: (1) OS refers to the outermost surface of Au and AAO at the opposite side. Because FE didn't attach to the outermost layer of AAO, the OS only refers to the outermost layer of Au in the present work

(Fig. S2). (2) Inner wall (IW) refers to the residual surface of the nanochannels except for the OS (Fig. 2a). We reduced the diameter of nanochannels from 25 ± 5 nm to 11 ± 3 nm by prolonging deposition time (0.1 nm/s for 2000 s) on purpose of restraining FE from entering the IW of nanochannels through the threshold effect (meaning that the stacking FE initially at the opening of nanochannels excluded the subsequent FE from entering IW to a great extent) (Fig. 2d, e and S3–S6), differing from the relatively free diffusion in nanochannel within larger channel diameter[23,26–29]. Then, the three different FE, including polyacrylic acid (PAA, Mw ~5000) through Van der Waals' force, poly(ethylene imine) (PEI, Mw ~10,000) through Van der Waals' force[30] and DNA (Mw ~11,000) through Au-thiol interactions[22,23] respectively, were attached to the OS of none@OS (Figs. S7 and S8). The as-obtained nanochannel-systems were named as PAA@OS, PEI@OS, and DNA@OS, respectively.

**Definitions of FE$_{OS}$ in nanochannel systems.** We further defined FE$_{OS}$ in the present nanochannel-system. Frankly, limited by current technologies, we cannot attach FE purely at the OS but not at the IW (Fig. 2f). Therefore, the depth of FE at the IW of the nanochannels was measured to be as much as 180 nm (PAA@OS), 75 nm (PEI@OS), and 23 nm (DNA@OS), using the ToF-SIMS and the scanning electron microscope (SEM)[23], which occupied a small part of the total depth of IW (65 μm) as 0.28%, 0.12%, and 0.04%, respectively. (Fig. 2g and S9). Compared to the distribution percentage of FE in the 1st stage through random indraft (≈100%)[26,27,29] and in the 2nd stage through Au–S interaction (5–30%)[23], the depth of FE occupied a tiny percentage (<0.3%) of the total depth of IW in this work, which could be attributed to the threshold effect during the penetration process of FE. We found that the FE depth at IW decreased with their molar mass that manifested the threshold effect (right figure in Fig. 2g, iv). We, therefore, defined the FE$_{OS}$ in this work (Fig. 2f), consisting of the FE at the OS of Au side and the FE at the IW existing near the opening of the nanochannels (Fig. 2g and S10).

**Explicit role of FE$_{OS}$ in regulating-ion-transport.** We then investigated the role of FE$_{OS}$ in the regulating-ion-transport in the nanochannel-system using a two-electrode cell with two symmetric Ag/AgCl electrodes and 0.1 M KCl solution (Fig. 3a). The current–voltage (I–V) plots of none@OS is asymmetric (Fig. 3b) owing to the asymmetric structure and surface properties (Figs. S5 and S6), classified as ion-current rectification (ICR) behavior, in which $I_{+2V}$ (current at +2 V) / $I_{-2V}$ (current at −2 V) is defined as ICR ratio ($f_{rec}$)[31]. After attached FE$_{OS}$, the $f_{rec}$ increased (for PAA@OS and DNA@OS) and the $f_{rec}$ decrease lower than 1 with an opposite polarity (for PEI@OS) (Fig. 3f). The variation of ion current above shows a similar trend as the previous reports (FE with distributions in the 1st stage described in Fig. 1)[30]. This effect was ascribed to the enhancement of negative charge at outer surface from highly negatived PAA rich in hydroxyl in PAA@OS or DNA with phosphodiester skeleton in DNA@OS (Fig. S11), or charge reversal of surface charge for the highly positive PEI@OS rich in amino, leading to the enhancement of ion accumulation and depletion (Fig. S16). In addition, the ion transport of nanochannels with independent FE$_{OS}$ was affected by ion strength (Fig. S13). Therefore, we estimated that the FE$_{OS}$ influenced the ion currents through nanochannel-system above independently.

In the 1st and 2nd stage, the numerical simulations coupled the Poisson–Nernst–Planck (PNP) equations[32] among the classical equations with steady state continuity equations were performed for FE$_{IW}$[16,17,21,28–30,33]. However, the three parameters of FE$_{IW}$

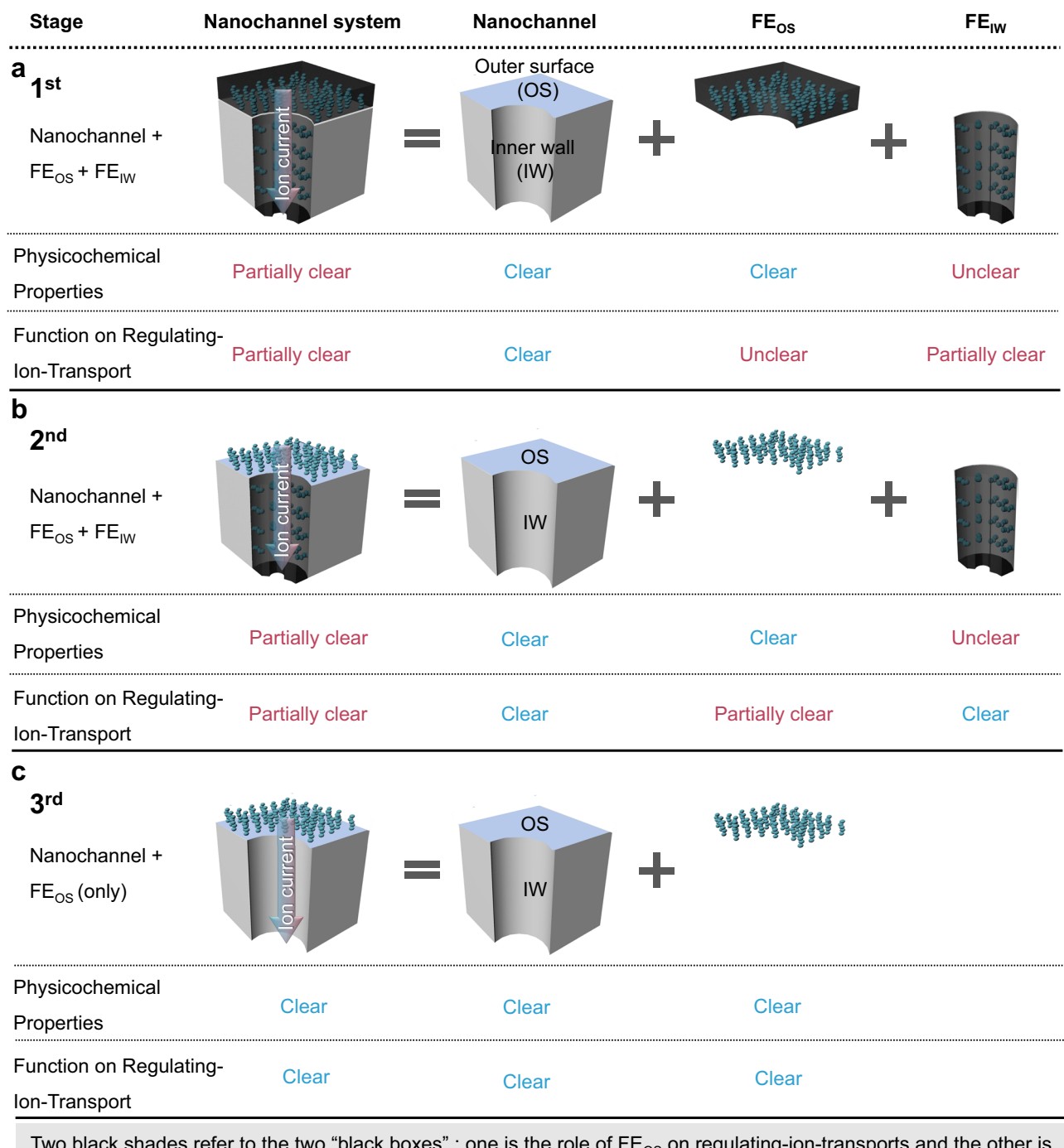

| Stage | Nanochannel system | Nanochannel | FE$_{OS}$ | FE$_{IW}$ |
|---|---|---|---|---|
| **a 1st** Nanochannel + FE$_{OS}$ + FE$_{IW}$ | = | Outer surface (OS) / Inner wall (IW) | + | + |
| Physicochemical Properties | Partially clear | Clear | Clear | Unclear |
| Function on Regulating-Ion-Transport | Partially clear | Clear | Unclear | Partially clear |
| **b 2nd** Nanochannel + FE$_{OS}$ + FE$_{IW}$ | = | OS / IW | + | + |
| Physicochemical Properties | Partially clear | Clear | Clear | Unclear |
| Function on Regulating-Ion-Transport | Partially clear | Clear | Partially clear | Clear |
| **c 3rd** Nanochannel + FE$_{OS}$ (only) | = | OS / IW | + | |
| Physicochemical Properties | Clear | Clear | Clear | |
| Function on Regulating-Ion-Transport | Clear | Clear | Clear | |

Two black shades refer to the two "black boxes" : one is the role of FE$_{OS}$ on regulating-ion-transports and the other is unclear physicochemical properties of FE$_{IW}$.

**Fig. 1 Designed nanochannel-systems attached with FE$_{OS}$ and FE$_{IW}$. a** Stage 1, FE$_{OS}$ and FE$_{IW}$ immobilized as a whole, in which the role of FE$_{OS}$ on ion transport and the properties of FE$_{IW}$ are inexplicit (two "black boxes" exist). **b** Stage 2, FE$_{OS}$ and FE$_{IW}$ as distinct part, in which the role of FE$_{OS}$ on ion transport began to be paid attention and investigated, but the properties of FE$_{IW}$ are still inexplicit (start to open the 1st "black box"). **c** Stage 3, in this work, independent FE$_{OS}$ without FE$_{IW}$ in nanochannel-system for regulating-ion-transport (further reveal the 1st "black box"), in which the properties of both FE$_{OS}$ and nanochannels are measurable, making the properties of the whole nanochannel-system explicit to a great extent (avoid the trouble from the 2nd "black box").

used in equation were unmeasurable, as (1) the depth of FE at IW, (2) the surface charge, and (3) the diameter of the nanochannels after attaching FE$_{IW}$. Hence, the hypotheses of the above three parameters were generated in the 1st and 2nd stage, which is unavoidable: (1) FE may not completely cover IW, but leaving the blank area of the deep IW hard for FE to reach[34]. (2) The surface charge of FE$_{IW}$ is substituted by the measurable surface charge of FE$_{OS}$. Surface charge density is usually different between FE$_{OS}$ and FE$_{IW}$, due to their different grafting densities[22]. Sometimes the charge of FE$_{OS}$ and FE$_{IW}$ inversed caused by the local polarization[35]. (3) The decrement of nanochannel diameter is roughly estimated by subtracting the straighten length of FE ($D - 2L_{SM}$, where $D$ represent the diameter of nanochannels and $L_{SM}$ represent the straighten

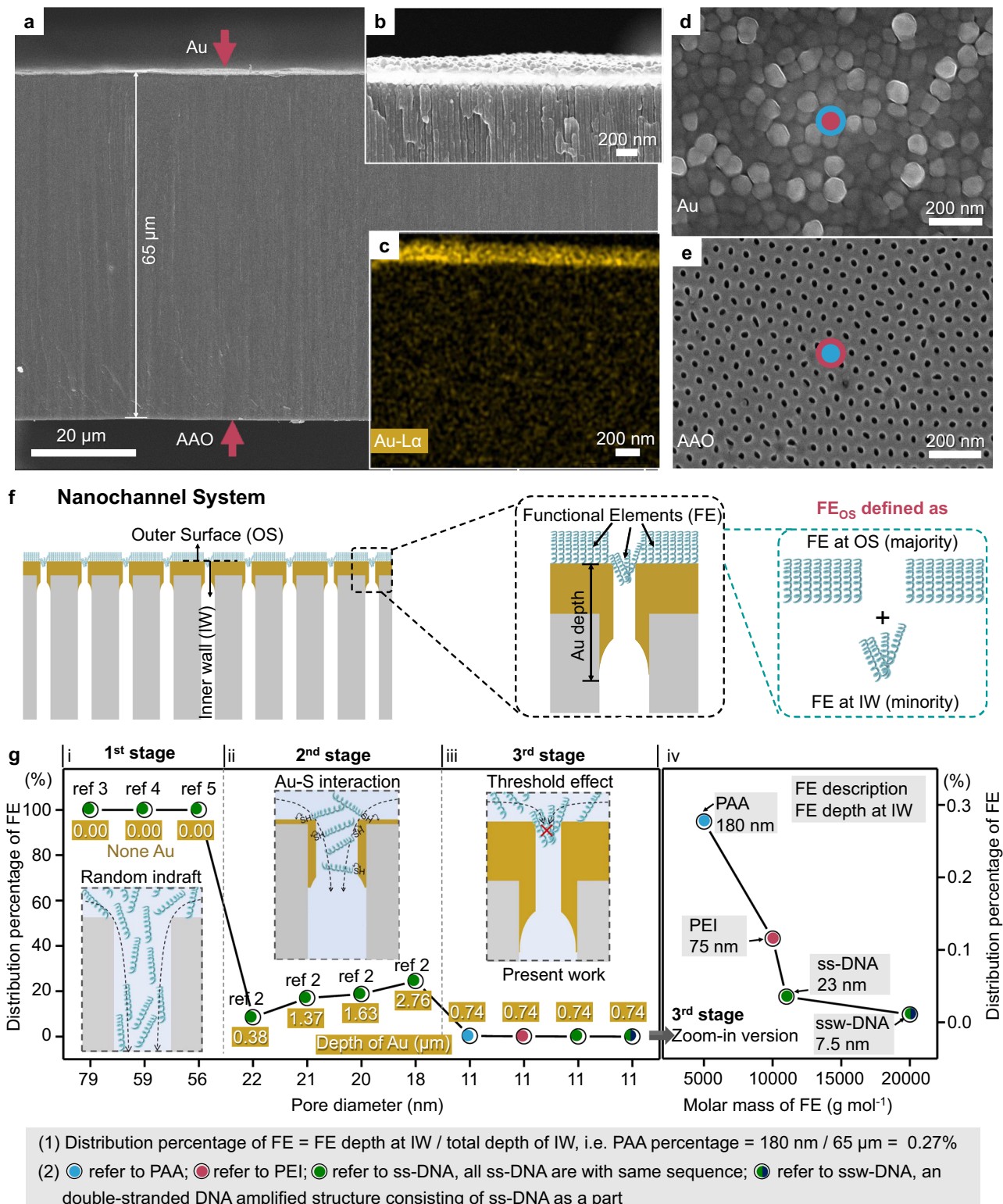

(1) Distribution percentage of FE = FE depth at IW / total depth of IW, i.e. PAA percentage = 180 nm / 65 μm = 0.27%

(2) ⬤ refer to PAA; ⬤ refer to PEI; ⬤ refer to ss-DNA, all ss-DNA are with same sequence; ⬤ refer to ssw-DNA, an
    double-stranded DNA amplified structure consisting of ss-DNA as a part

length of FE)[36]. However, the FE are mostly not straight, i.e.,
single-strand DNA[25]. The results from the numerical simulations,
therefore, deviate from experiments and even have randomness,
which indicates that the above hypotheses bring the deviations or
even sometimes errors. In our nanochannel-system, we measured
(1) the depth of the part of $FE_{OS}$, (2) the surface charge, and (3)
the diameter of nanochannels after attaching $FE_{OS}$ using ToF-
SIMS, SSZPA, and AFM[37], respectively (Figs. 2g, 3d, 3e and Figs.

S11, S12). Both the qualitative and the quantitative variation of
ICR behavior from numerical simulations fitted well with the
experimental results, which indicated the explicit regulating-
ion-transport accomplished in 3rd stage nanochannel-system
(Fig. 3f, g).

One of the important features of FE is of versatile physical and
chemical properties, nanochannel-system adapt to a broad range
of applications spanning from osmotic energy conversion

**Fig. 2 The characterization of none@OS and the FE$_{OS}$. a** SEM image of none@OS from sectional view. The thickness of nanochannels is 65 μm. **b** Zoom-in version of Au coating side. **c** The corresponding energy dispersive X-ray spectroscopy of **b**. **d**, **e** SEM images of the OS coated with Au (**d**) and without Au (**e**) of none@OS from top view. **f** A scheme showing the present nanochannel-system and the FE distribution near the opening of nanochannel-system. The exposed surface to FE in none@OS includes OS and IW, of which the whole OS and a tiny fraction of IW are attached with FE herein. The FE$_{OS}$ in this work consists of all FE$_{OS}$ and a very small amount of FE$_{IW}$. **g** Comparison of the distribution percentage of FE at the IW in the (i) 1st, (ii) 2nd, and (iii) 3rd stage. In the 1st stage, the FE occupy the total depth of IW (≈100%) through the random indraft of FE (the inset)[26,27,29]. In the 2nd stage, the distribution percentage of FE at IW decrease down to 5–30% through the Au–S interaction between thiol-modified FE and IW (the inset)[23]. In the 3rd stage, the distribution percentage of FE at IW sharply decline near zero. In the (iii) zoom-in version (iv), the distribution percentage of FE decrease with their molar mass, which demonstrates the threshold effects in the 3rd stage.

devices[15–17,33,38] to biosensors[26,27,39–41]. Meanwhile, a nanochannel system combining with electrochemistry in a confined space is now a crucial promising field[42,43], which is utilized to dynamically monitor the single molecule[44], understand the chemical reaction[45], characterize the single particle[46], and probe single living cell[47], etc. Here, we investigated whether the nanochannel-system in 3rd stage with FE$_{OS}$ explicit regulating-ion-transport could fulfill the applications above or even with special performances to the 1st and 2nd.

**Impact from FE$_{OS}$ on osmotic energy conversion devices**. The osmotic energy conversion devices were fabricated using nanochannels with only FE$_{OS}$ without FE$_{IW}$ (Fig. 4a–d and Figs. S16–S19)[15–17]. The output power density of nanochannel system with independent FE$_{OS}$ was estimated according to the equation $P_L = I^2/R_L$, where $I$ is the current across the circuit and $R_L$ is the external load resistance (Fig. S21). It was found that (1) output max power density increased with PAA concentration owing to the enhanced ion selectivity (Fig. 4e and Fig. S21); (2) $R_{channel}$ was almost unchanged with increase of concentration PAA (Fig. 4e and Fig. S21), where $R_{channel}$ was the internal resistance of the nanochannels. For (1), it is easy to understand the selectivity of the cell increased with the negative-charged FE$_{OS}$ (PAA)[20]. For (2), we speculated the $R_{channel}$ did not obviously change with the FE$_{OS}$. In order to verify our assumptions, two FE$_{OS}$ layer by layer (LbL) assembled on the OS of nanochannels (PAA and PEI sequentially in Fig. 4f–h and Figs. S22–S24; and PEI and PAA sequentially in Figs. S25 and S26)[30,48]. No distinctly increase of $R_{channel}$ was observed during two FE$_{OS}$ sequential assembly at the OS in the 3rd stage (Fig. 4i) owing to the ignored resistance induced by FE$_{OS}$, while $R_{channel}$ increased with the LbL assembly of PAA and PEI at the IW in the 1st and 2nd stages (Fig. S27). The above results demonstrated that the increase of FE$_{OS}$ (with selectivity) enhanced the "output max power density" of osmotic energy conversion devices (3rd stage), without obviously raising $R_{channel}$, which indicated a new conceptual route to design large power density nanochannel-system[27,49–51].

**Impact from FE$_{OS}$ on biosensors**. We further demonstrated a sensing strategy employing single-strand DNA probes as FE$_{OS}$ in the nanochannel-systems (3rd stage) (Fig. 5a), for the detection of a broad range of targets including inorganic ions (Hg$^{2+}$ with 1 pM limit of detection, LoD), small molecules (ATP with 1 pM LoD), proteins (lysozyme with 1 pM LoD), and cancer cells (MCF-7 cells with 400 cells mL$^{-1}$ LoD) (Fig. 5b, c and Fig. S28). The targets were specifically captured by the designed FE$_{OS}$ as probe and tailored the surface charge of the OS locally, which affect the asymmetry of surface potential in between OS and IW and change the ion transport in form of $f_{rec}$ signal. To confirm the sensing mechanism above, we took ATP detection using ssw-DNA as an example. In ssw-DNA structure, one kind of repeating units in ssw-DNA was designed as ATP aptamer, which specifically bonded with ATP and caused the disassembly of ssw-DNA

(Fig. 5d–g and Fig. S29). The variation of surface potential under the disassembly of ssw-DNA triggered by different concentration ATP was quantitatively characterized through the electrochemical approaches (Fig. 5h and Fig. S30). The mechanism was also confirmed by the change grafting density of DNA at OS (Fig. S31). The selective detection for ATP was also realized based on surface-charge-response sensing mechanism (Fig. 5h, i). In the 1st and 2nd stage, because probes (as FE$_{IW}$) were immobilized at the IW, a confined space usually with diameter <100 nm, the targets with a size beyond the diameter of nanochannels can't sufficiently contact with probe and efficiently recognized. In the 3rd stage, the OS possess the receptive characteristic for probes or targets with the size beyond the diameter of nanochannels. The recognition between probes and targets took place at the OS, which is relatively more free-spaces compared with IW in nanochannels. Here, we realized a nearly "universal" biosensor approach according to the two successful sensing processes: one is the DNA amplifications (ssw-DNA) as the probes, whose diameter is larger than the diameter of the nanochannel-system (some probes is up to 30-fold larger in Fig. 5e–g), and the other is the MCF-7 cells as the targets, whose diameters is about than 2 magnitudes larger than the diameters of the nanochannel-system.

**Discussion**
In conclusion, we have built a de novo designed "minimalist" nanochannel-system with an explicit regulating-ion-transport feature, which has been achieved using FE$_{OS}$ independently without FE$_{IW}$. The troubles from the "black box" for the properties of FE$_{IW}$ in the 1st and 2nd stage are well avoided without using FE$_{IW}$, meanwhile the other "black box", the role of FE$_{OS}$ on regulating-ion-transport, are further demonstrated from the "coadjutant" of FE$_{IW}$ in the 2nd stage to the "monodrama" in the 3rd stage. The use of independent FE$_{OS}$ is a new attempt to separate the enrichment, screening, and recognition process taking place at OS from the ion transport process taking place at IW within a confined space at the nanoscale. Furthermore, utilization of independent FE$_{OS}$ under a less restricted environment than FE$_{IW}$ will endow nanochannel-systems with (1) more measurable properties, such as hydrophobicity (the contact angle measurement is available for FE$_{OS}$, which is hardly achieved for FE$_{IW}$ in nanoscale confined space) (Fig. S32), (2) more characteristic technologies, such as X-ray photoelectron spectroscopy (XPS), ion microprobe mass analysis (IMMA) (the detection on FE$_{IW}$ requiring for destructive treatment, such as ion thinning, but not necessary for FE$_{OS}$), (3) more new performances limited to diffusions in nanochannels previously, such as reusage of FE (Fig. S33), rapid responses to targets (Fig. S34). The FE$_{OS}$ in 3rd stage like "ignition system" trigger the regulatable substrate transports through nanochannels and the abroad application space of nanochannel-systems.

**Methods**
**Materials**. Poly (acrylic acid) (PAA, MW ~5000) was purchased from Ryon Biological Technology (Shanghai, China). Polyethyleneimine (PEI, MW ~10,000)

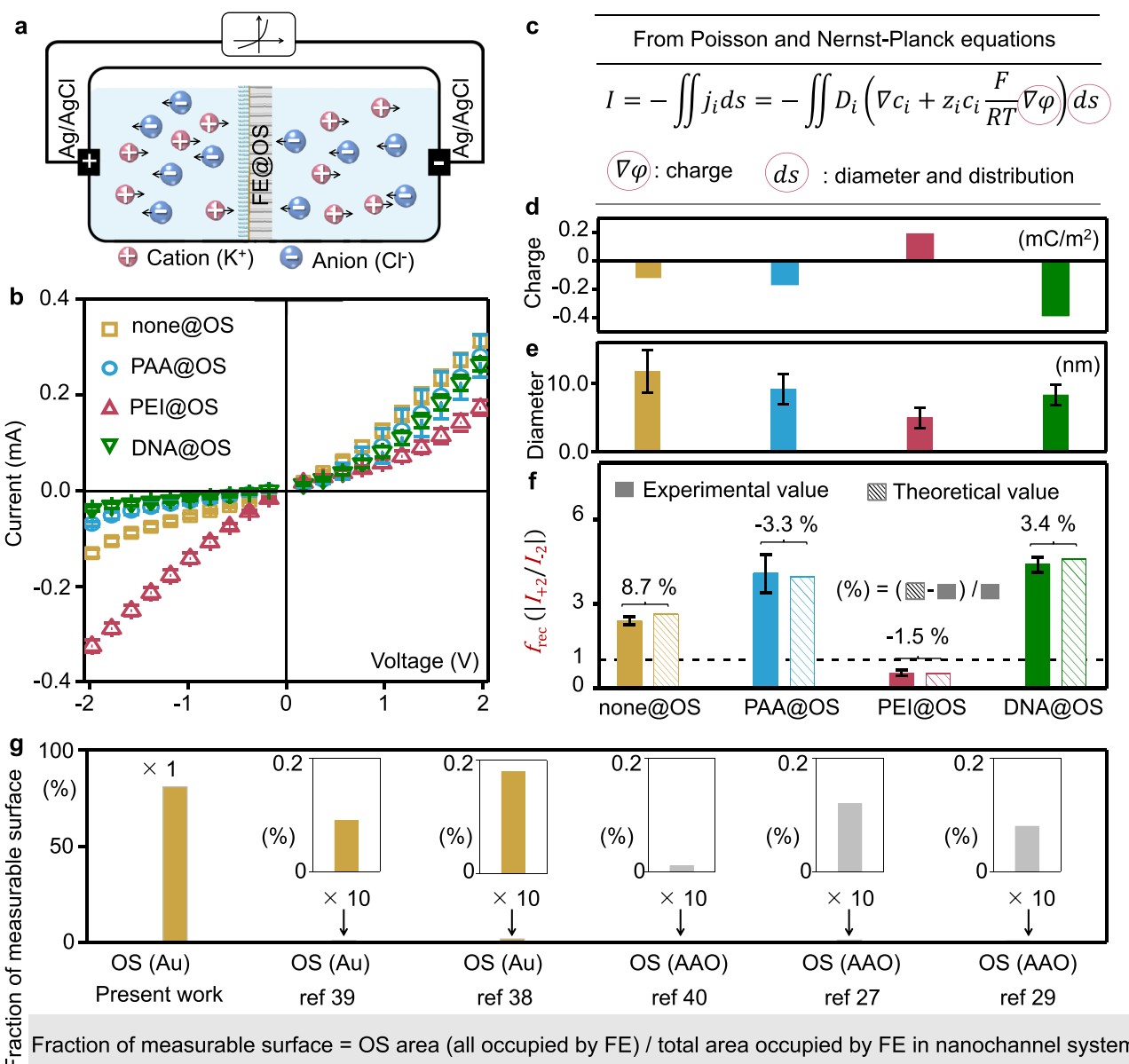

**Fig. 3 Explicit role of FE$_{OS}$ in regulating-ion-transport. a** Scheme of a two-electrode cell. **b** I–V curves characterizing the ion transport through nanochannel-system. **c** The PNP equation was used for numerical simulations of the effect from FE$_{OS}$ on the ion transport through nanochannel, where two variables remain: $\nabla c_i$ and $\nabla \varphi$ valued by the surface charge density ($\sigma$) (measured by SSZPA, Fig. 3d) (1), the diameter of nanochannels (measured by AFM, Fig. 3e) (2) and the depth of FE at IW (measured by ToF-SIMS, Fig. 2g) (3) (More details in the section "Numerical simulation" in "Method" section). Thus, unless otherwise stated, the I is available from numerical simulations using the three parameters (1), (2) and (3) measurable in the nanochannel-system with FE$_{OS}$ only but without FE$_{IW}$. **f** Comparison of the rectification ratio ($f_{rec}$) measured from I–V tests (experimental value) with the $f_{rec}$ from numerical simulations based on classical equations using measured parameters (theoretical value), but not estimated values like the previous works in 1st and 2nd stage. **g** Fraction of the measurable surface, which is the area proportion of the OS of measurable properties using mostly current test technologies, in the total surface occupied by FE (Details in Table S1). Data from the present work and the previous works using functional solid-state nanochannels[27,29,56–58]. For I–V tests, five chips were used to obtain each error bar. Statistics of diameters have been done by counting 100 nanochannels for each kind of nanochannels.

was purchased from Damas-beta. Tris (hydroxymethyl) aminomethane (Tris) was purchased from Alfa Aesar. KCl, NaCl, and MgCl$_2$ were obtained from Aladdin reagent (Shanghai, China). Adenosine 5′-triphosphate (ATP) disodium salt solution, uridine Triphosphate (UTP), cytidine triphosphate (CTP) and guanosine triphosphate (GTP) were purchased from Sigma-Aldrich. HeLa and PC3 cells were obtained from Chinese Center for Typical Culture Collection (Wuhan, China) and cultured in DMEM (Gibco) supplemented with 10% (v/v) fetal bovine serum (FBS), 2 mg/mL NaHCO$_3$, and 100 U/mL antibiotics 15 (penicillin-streptomycin). MCF-7 cells were purchased from KeyGEN Biotech Co. Ltd. (Nanjing, China) and cultured in RPMI-1640 (Gibco) supplemented with 10% FBS, 2 mg/mL NaHCO$_3$, and 100 U/mL antibiotics (penicillin–streptomycin) at 37 °C in a humidified 5% CO$_2$

atmosphere. AAO membranes were purchased from Pu-Yuan Nano Technology Co. Ltd. (Hefei, China). The thickness of AAO membrane is 65 µm. All solutions were prepared using Millipore Milli-Q water (18 MΩ cm). All oligonucleotides are synthesized and purified by Sango Biotech Co. Ltd, (Shanghai, China). The sequences are shown in Tables S2 and S3.

**Fabrication of nanochannel systems**. Preparation of none@OS. A physical vapor deposition (PVD) method was employed using AE Nexdap PVD platform (Angstrom Engineering Inc.) to prepare nanochannels with independent OS[22,23]. Two kinds of depositing materials as Au and Cr were used. To enhance the stability of

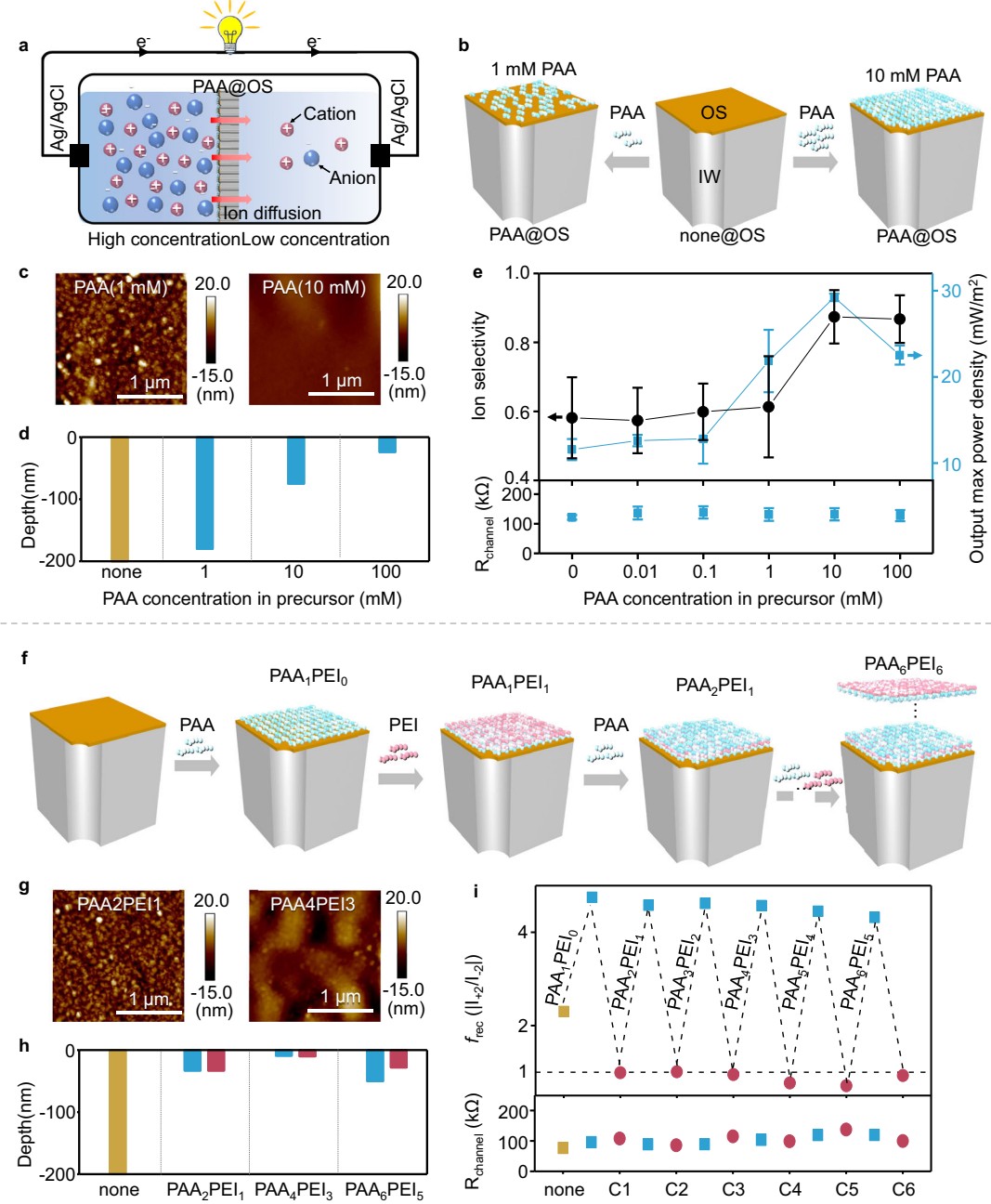

**Fig. 4 Effect from FE$_{OS}$ on osmotic energy conversion devices. a** A scheme showing the working mechanism of the osmotic energy conversion devices using nanochannel-system. The electricity by reverse electrodialysis is generated under salt gradient using a nanochannel system containing only FE$_{OS}$. **b** Fabrication of the nanochannel-system attached with PAA as FE$_{OS}$ (PAA@OS) using precursor with different PAA concentration (1 or 10 mM). **c** AFM of PAA@OS using 1 and 10 mM PAA in precursor. **d** Depth distribution of the part of FE$_{OS}$ in nanochannel-system using precursor solution with different PAA concentration from ToF-SIMS. **e** Output max power density, ion selectivity and $R_{channel}$ of PAA@OS with different PAA concentration in precursor. **f** Scheme showing LbL assembly of PAA and PEI at OS sequentially. **g** AFM of the OS after LbL assembly. **h** Depth distribution of the part of FE$_{OS}$ (PAA and PEI) in nanochannel-system after LbL assembly. **i** Reversal $f_{rec}$ through the LbL assembly of PAA and PEI and corresponding $R_{channel}$. The numerical simulations of two samples above, PAA$_4$PEI$_3$@OS and PAA$_4$PEI$_4$@OS, using their measured parameters were performed, which well fit the experimental results (Fig. S24). For the osmotic energy conversion devices, five devices using FE@OS were assembled to obtain each error bar of power density and $R_{channel}$.

Au, one layer of Cr with a thickness of 10 nm was first deposited at the one side of AAO. The Au was followed deposited at the same side of Cr layer. The circular targets were approximately parallel to the AAO membranes, ensuring the deposition direction perpendicular to the membranes. The deposition was taken by the successive deposition without replacing the target materials or releasing vacuum. The successive deposition ensured no secondary pollution at the first deposited layer. The low depositing speed of Au was applied as 0.1 nm s$^{-1}$ and the time was 2000 s. The depositing speed was calibrated by the deposited thickness on the surface of the flat silicon wafer at nanometer level. The as-synthesized sample was named none@OS.

Preparation of PAA@OS, PEI@OS, and DNA@OS. Aqueous solution including PAA or PEI with different concentrations was prepared. Before attaching PAA or PEI, the Au-coated OS in none@OS was treated by Ar plasma for 60 s to remove impurities. Then, the PAA solution or the PEI solution was dip-coated at the OS. The dip-coating continued for 2 min. Then the sample was washed throughout by water, and dried under N$_2$ gas. The sample was named PAA@OS and PEI@OS, respectively. Thiol-modified DNA was attached at the OS of none@OS using similar dip-coating method. The reaction time continued for 60 min and the ample was named DNA@OS. The as-prepared samples were then applied for further measurement and characterizations.

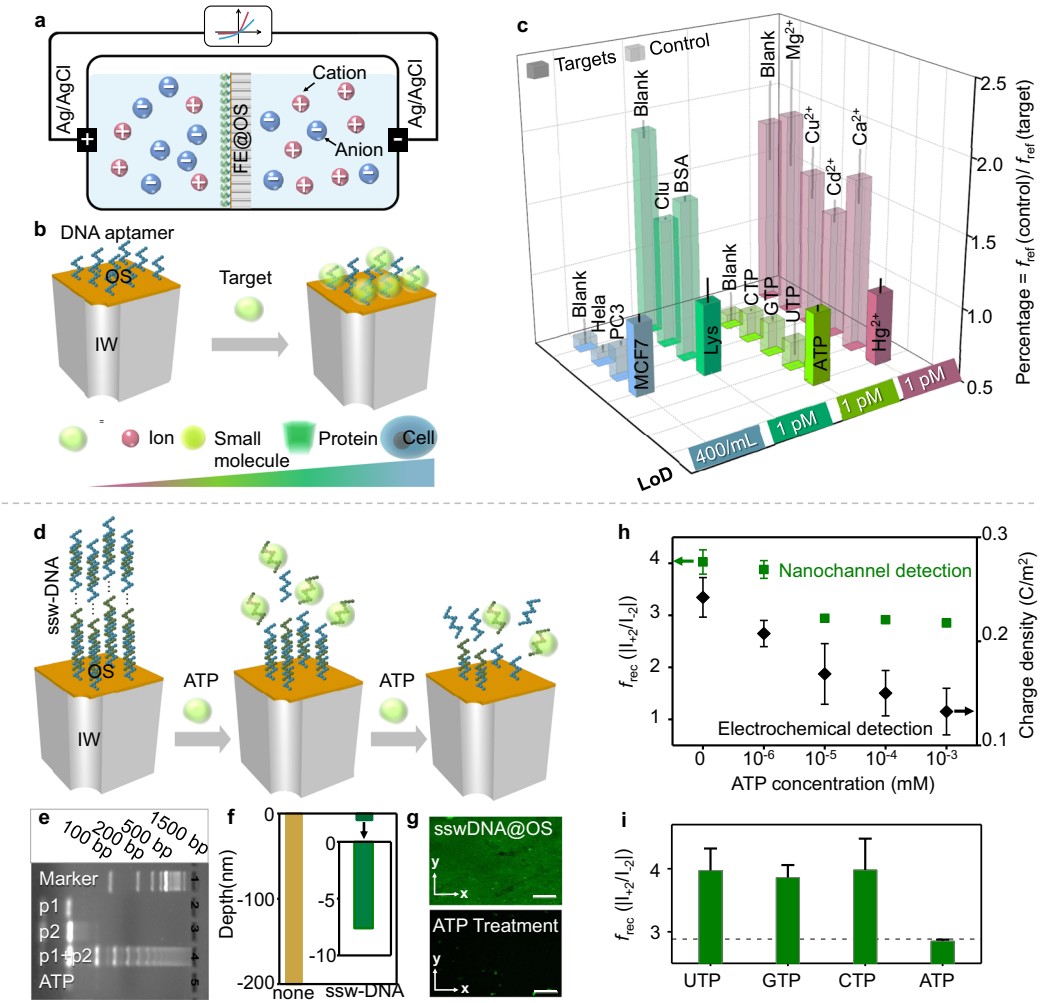

**Fig. 5 Sensing performances of FE$_{OS}$ as probes. a** A scheme showing the working mechanism of the sensor using FE$_{OS}$ as probes. **b** Capture process of multi-scale targets through designed single-stand DNA. **c** Sensitivity and selectivity of the DNA@OS (DNA is a designed sequence specifically bonding with targets) for the recognition of ions (Hg$^{2+}$), small molecules (ATP), protein (Lysozyme) and cells (MCF7). The selective detection of multiscale targets using FE$_{OS}$ was realized based on the change of $f_{rec}$ signal output induced by the surface charge at outer surface. LoD is defined as the limitation of detection for the targets. **d** Formation of "supersandwich" DNA structure (ssw-DNA) with long concatamers through the successive hybridization of alternating DNA unit. And gradual disassembly of ssw-DNA based on interaction between ATP and the repeat DNA units through increasing ATP concentration. **e** Agarose gel electrophoresis characterizing of the ssw-DNA: 1) DNA marker; 2) p1; 3) p2 (ATP ampter); 4) p1 + p2; 5) target+p1 + p2. **f** Depth distribution of the "supersandwich" DNA in nanochannel-system using ToF-SIMS. **g** Laser scanning confocal microscopy of the OS after the assembly (top) and the disassembly of ssw-DNA (bottom). The scale bar is 20 μm. **h** The LoD of ATP using ssw-DNA as probe based on nanochannel method and electrochemical method (Fig. S30). **i** Specificity of ssw-DNA@OS for ATP, in contrast with other NTPs. For the sensing performances part, five sensors using FE@OS were established to obtain each error bar of sensitivity and specificity.

LbL self-assembly of charged polyelectrolyte. For LbL self-assembly of PAA and PEI on the outer surface of AAO/Au nanochannels, the process was described as follows: an aqueous solution containing PAA (1 mM) was firstly spread onto at the OS of none@OS for the adsorption of PAA polyelectrolyte, and the modification time was 2 min. After the adsorption of first layer of PAA, the solution of PEI was then spread on the same side of none@OS. Subsequently, other alternative layers of PAA/PEI were also electrostatically deposited on the OS of none@OS using the same procedure.

Assembly and disassembly of ssw-DNA. The DNA supersandwich structure (ssw-DNA) was prepared by mixing the thiol-modified cp-DNA solution (1 μM) into the mixture of equimolar P1 and P2 (1 μM) for 1 h. Then the solution containing ssw-DNA was dropped onto the OS of none@OS, and the reaction time was 60 min, named ssw-DNA@OS. The disassembly of ssw-DNA was achieved by treating the ssw-DNA@OS with different concentrations of ATP (1 nM–1 μM) for 60 min.

**Detection of Hg$^{2+}$, ATP, lysozyme, and MCF-7 cells**. The aptamer solution (1 μM) for different targets were dropped onto the OS of none@OS at room temperature for 60 min. After that, 6-mercapto-1-hexanol (MCH, 100 nM) was

added onto the OS for 60 min to prevent nonspecific adsorption. Subsequently, the aptamer modified nanochannels was treatment by the suspension containing various targets. The treatment time was 60 min. After the interaction, the sample was washed by buffer to remove the nonspecifically adsorbed targets. And then the as-treated samples were then applied for characterizations.

Cells were first cultured in flasks with Dubelcco's Modified Eagle's Medium (DMEM) at 37 °C under 5% CO$_2$ in the cell incubator followed by centrifuging at $600 \times g$ for 5 min, and re-dispersed in PBS (10 mM, pH 7.4) with a density of $5 \times 10^5$ cells mL$^{-1}$. A certain amount of cells suspension reacted with aptamer modified nanochannels for 60 min. After cells capture, the sample was washed by PBS buffer to remove the nonspecifically adsorbed cells. And then the as-prepared membranes were then applied for characterizations.

**Characterization**. SEM images and EDX analyses were taken with a field-emission scanning electron microscope (SU8010, Hitachi, Japan) equipped with Energy Dispersive Xray spectroscopy (EDS, BRUKER AXS, Germany). All samples were coated with carbon (5 nm) prior to SEM examinations. Secondary ion mass spectra of as-prepared samples were characterized by ToF-SIMS V (IONTOF, GmbH). A Bi liquid metal primary ion source was applied with an angle of 45° relative to the

sample surface with a pulsed $Bi^{3++}$ primary ion beam of 30 keV and shave off fresh 60 μm × 60 μm areas for each analysis. The ToF analyzer was installed at an angle of 90° to the sample surface. Negative secondary ion spectra were collected. Mass calibration was carried out using standard procedures (mass resolving power >5000). The properties of OS of nanochannel system were characterized by an atomic force microscope (Multimode 8, Burke). The determination of OS's surface zeta potential of nanochannel system were carried out by using SSZPA (SurPASS, Anton Paar Ltd., Austria). The pH dependence of the zeta potential and the iso-electric point (IEP) were obtained in a 0.1 M potassium chloride at the same electrical conductivity. The fluorescent measurement of nanochannel functiona-lized with fluorescent-dyed DNA at OS were performed by using laser scanning confocal microscope (LSM 880 confocal microscopy, Carl Zeiss) equipped with a FemtoSecond Laser (Coherent Inc.) The fluorescent-dyed membranes were placed at cover glass filling with water (around 1 cm² membrane with 20 μL water). For the analysis of assembly and dissembly process of ssw-DNA, the mixture was analyzed and observed using native PAGE gel. The gel was run in 16% acrylamide (containing 19/1 acrylamide/bisacrylamide) solution with 1× TBE buffer, at 100 V constant voltage for 1.5 h. The gel was directly imaged or stained with GelRed (Biotium) for 20 min to image the DNA position by using Tanon imaging system (Tanon 5200 Multi). The water contact angle measurements were operated on a DSA100 contact angle analyzer at ambient temperature and around 30% humidity. A drop of water (5.0 μL) was dropped onto the surfaces of nanochannel membranes.

The surface charge density for the OS with ssw-DNA grafting. The surface charge density with ssw-DNA grafting were tested using the system described in Fig. S28 and calculated based on the following equation[52]:

$$Q = \frac{2nFAD_0^{1/2}C_0}{\pi^{1/2}} t^{1/2} + Q_{dl} + nFA\Gamma_0,$$ (1)

where $n$ is electron transfer number, F is the Faraday constant, $A$ is the electrode area (cm²), $D_0$ is the diffusion coefficient (cm² s⁻¹), $C_0$ is the bulk concentration (mol cm⁻²), $Q_{dl}$ is the capacitive charge (C), and $nFA\Gamma_0$ is the charge from the reduction of $\Gamma_0$ (mol cm⁻²) of adsorbed redox marker.

The samples were mounted in between the two halves of a homemade electrochemical cell, which contains 1.0 mL KCl solution in each cell. I–V plots were recorded by an electrochemical workstation (CHI, ShangHai) and Keithley 6487 picoammeter (Keithley Instruments, Cleveland, OH). Two Ag/AgCl electrodes were used to apply the potential. The working electrode was placed on the Au side and the reference electrode on the AAO side if not particularly mentioned. The effective area of the membrane for ion conduction test is about 7 mm². All measurements were performed at room temperature, and each test was repeated for five times. Five membranes at least were used to obtain average values. To study the ionic rectification properties, a scanning voltage from −2.0 to 2.0 V was applied across the membrane. The ion concentration on the Au side and the AAO side was kept equal (0.1 M KCl, pH 7.0), if not specially mentioned. To study the energy conversion properties[53] a salt gradient was designed, where the concentration of the solution on the Au side is higher than that on the AAO side if not particularly mentioned.

**Numerical simulations.** The ionic rectification phenomenon was theoretically investigated using a commercial finite-element software package COMSOL Multiphysics[54,55].

$$J_i = D_i \left( \nabla c_i + \frac{z_i F c_i}{RT} \nabla \varphi \right),$$ (2)

$$\nabla^2 \varphi = -\frac{F}{\varepsilon} \sum z_i c_i,$$ (3)

$$\nabla \cdot J_i = 0.$$ (4)

Here, Eq. (2) is the Nernst–Planck equation that descripts the transport properties of a charge nanochannel and Eq. (3) is the Poisson equation that descripts the relationship between the electric potential and ion concentration in the nanochannels. The model is generally simplified using steady-state conditions (Eq. (4)). The electroosmotic flow was neglected in this work. The coupled Eqs. (2–4) can be solved utilizing appropriate boundary conditions (Eqs. 4, 6 and Table S4). The solution yields the concentration field $c_i$ for all species and the potential $\varphi$ distributions in the nanochannels.

$$\mathbf{n} \cdot \nabla \varphi = -\frac{\sigma}{\varepsilon},$$ (5)

$$\mathbf{n} \cdot J_i = 0.$$ (6)

The total ionic current through the nanochannel can be calculated:

$$I = \iint J_i ds = -\iint D \left( \nabla c_i + z_i c_i \frac{F}{RT} \nabla \varphi \right) ds.$$ (7)

The formula above was used for numerical simulations of the effect from FE$_{OS}$ on the ion transport through nanochannel, where $i$ refer to cation ($K^+$) and anion ($Cl^-$) and $J_i$ refers to ionic flux[16]. When the properties of electrolyte are set, the invariants include: $D_i$ as the diffusion coefficient, $c_i$ as the concentration of $K^+$ and

$Cl^-$, $z_i$ as the valence of $K^+$ and $Cl^-$, $T$ as the thermodynamic temperature. Meanwhile, in this formula, the constants include: $F$ as the Faraday's constant and $R$ as the ideal gas constant. As a result, two variables remain: $\nabla c_i$ describes the concentration gradient of ion along nanochannels under external electric field and $\nabla \varphi$ describes the electrical potential gradient along nanochannels, in which the $\varphi$ is valued by the surface charge density ($\sigma$) of the FE$_{OS}$ and the nanochannels (measured by SSZPA, Fig. 3d) (1). The $\nabla c_i$ and the $\nabla \varphi$ are calculated by integrating the $c_i$ of electrolyte and $\varphi$ over the cross-section area (d$s$). The integration of d$s$ is valued by the diameter of nanochannels (measured by AFM, Fig. 3e) (2), and the depth of FE at IW (measured by ToF-SIMS, Fig. 2g) (3).

## Data availability

The data that support the plots within this paper and other finding of this study are available from the corresponding author upon reasonable request.

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

## Acknowledgements

This work is supported by the National Natural Science Foundation of China (22090050, 21974126, 21874121, and 51803194). This research is supported by the Hubei Provincial Natural Science Foundation of China (2020CFA037), Zhejiang Provincial Natural Science Foundation of China under Grant No. LY19B030001 and LD21B050001. The project is supported by the Open-end Funds from Engineering Research Center of Nano-Geomaterials of Ministry of Education (NGM2019KF013) and the Fundamental Research Funds for National Universities, China University of Geosciences (Wuhan). This research work was supported by the Open Funds of the State Key Laboratory of Electroanalytical Chemistry (SKLEAC202003) and the National Key Research and Development Program of China (2018YFE0206900).

## Author contributions

F.X. directed the project. P.G. conceived and designed the experiments. Q.M. fabricated devices, performed measurements and carried out data analysis with help from Y.L., R.W., H.X., and Q. D. carried out the numerical simulations. P.G., Q.M., and F.X. wrote the manuscript. All authors contributed to discussions.

## Competing interests

The authors declare no competing interests.
