## [Peer Review File · Nature Communications]

REVIEWER COMMENTS

Reviewer #1 (Remarks to the Author):

The manuscript probes the importance of an outer surface of a porous membrane for ionic transport and sensing capabilities of a porous system. I do not believe this work belongs to Nature Communications; importance of the membrane surface and pore walls functionalities on transport in anodic alumina was reported before:

<https://www.nature.com/articles/s41467-017-02447-7>

<https://www.nature.com/articles/s41467-018-06873-z>

In addition, the manuscript presents a wide spectrum of experimental data with nearly no explanation. Figure 2 is incomprehensible. It is unclear how the data in Figure 4 was obtained; the Authors for example show values of ion selectivity and maximum power density without any information how this was obtained. It is not clear which advantage the presented system even has over previously reported platforms used for similar sensing mechanisms.

I would suggest the Authors to expand the manuscript with explanation and comparative studies and submit it elsewhere. As mentioned above, a system with chemistries of the outer surface and pore walls controlled separately was reported before.

Reviewer #2 (Remarks to the Author):

The manuscript 'Towards Explicit Regulating-Ion-Transport: Nanochannels with only Function Elements at Outer-Surface' presented an interesting manuscript, in which the authors designed nanochannel-system to regulate the ion transport process by using FEOS independently without FEIW. They discuss the effect of outer surface on the ion transport process by using different chemical agents including PAA, PEI, and DNA molecules. The Finite Element Modeling simulation was also conducted to test the correlation of FEOS and i-V result. At the end, this unique nanopore array also was used to achieve osmotic energy conversion and biosensing. All the results in the manuscript show the advantages which the FEOS promote osmotic energy power density and can accommodate targets with size beyond the diameter of channels. The manuscript provides an important advance in ionic channel field, the experimental procedure is thorough with a high number of repeats. For these reasons, I fully support it to be published in Nat. Commun. That said, there were a couple aspects of the work which I felt required some further explanation, clarification or experimental evidence to ensure a reader fully understanding.

Below are a few points which may be worth thinking about:

1. In introduction part. The authors stated that it is necessary and difficult to study the effect of FEIW on ion transport. Whereas, the main purpose of this manuscript is the effect of FEOS rather than FEIW on ion transport. It seems that the authors overemphasize the role of FEIW in ion transport. Thus, it is suggested that the authors polish the introduction part and make the manuscript more readable.
2. The author mentions that PNP equations using the measured parameters of the FEOS, were well fitted with the experiment results in 3rd stage, indicating the explicit regulating-ion-transport accomplished by FEOS without FEIW. What is the advantage of ion-transport-regulation using FEOS only? Can the author list some applications or add more information about ion-transport with only Functions-Elements at Outer-Surface in the introduction?
3. Figure 1 depicts designed nanochannel-systems attached with FEOS and FEIW at stage1, 2 and stage 3. It is better to label in detail in the figure 1 about black box and the unclear properties or functions. For example, can the authors present the difference between 1st and 2nd. The meaning of 'Properties' and 'Functions' should be clarified, contributing to the understandable of this figure.
4. In 'Explicit role of FEOS on regulating-ion-transport' section, the author discussed the rectification

ratio (frec) measured from I-V tests. How about the ion transport dynamics with FROS only? It is better to illustrate the underlying reason in figure 2b and figure 2f about the correlation of rectification and chemical composition.

5. In figure 4a, the author gives a scheme showing the working mechanism of the osmotic energy conversion devices using nanochannel-system but the 'osmotic energy conversion' meaning is not obvious. Please add more detail to make us better understand. With the increasing PAA concentration in precursor, the ion selectivity and the output max power density both increase and has the same tendency. How about the relationship between them?

6. It is better to add more words about 'Impact from FEOS on biosensors' section to show that the author has realized a nearly 'universal' biosensor approach: one is the DNA amplifications as the probes and the other is the MCF7 cells as the targets. The mechanism of this biosensor should also be presented, why do the presence of target would trigger the changes in ion rectification ratio? Related FEM simulation is needed here to support the experimental result and speculation.

7. In figure 5c, this nanochannel-system can detect MCF7 cells. Could the author explain the reason why can this device accommodate targets with size beyond the diameter of nanochannels? Why does the percentage of MCF7 cells is larger than Hela cell and PC3 cell? The legend of figure 5c could be written more detailly.

8. The physical model and details (e.g. Geometrical Model, Boundary setting, and Mesh Parameters) of FEM simulation should be presented in supporting information. Furthermore, the related explanation for PNP equation also should be listed.

9. As the authors mentioned in the manuscript, the behavior of ion transport in a confined space is different from that in bulk solution. For example, the asymmetric geometry of confined space and asymmetric distribution of charge in confined space contribute to the phenomenon of ion current rectification. Besides, the electrochemically confined effect within the nanochannel or nanopore allows the single molecule sensing with high spatial and temporal resolution. Thus, I wonder can the author make comments on the electrochemistry in a confined space? And demonstrate the effect of confined effect on the ion transport process in detail.

10. There are also some format issues the authors should notice and check carefully:

1) The volume of journal should be bold in reference 43 and reference 44.

2) The journal name should be abbreviated as 'Angew. Chem. Int. Ed.' in reference 26.

Reviewer #3 (Remarks to the Author):

In this paper, the authors investigated the impact of functional elements at the outer surface (FE_OS) of nanopores on the regulation of ionic transport. By depositing Au on one side of the system, shrinkage of the pore exit is achieved, which limits the functional polymers to anchor to the inner surface of the nanochannel. As a result, the authors were able to focus on the role of FE_OS in the function of the nanochannel. The experiment showed that the FE_OS enhance the power density and ionic selectivity of the nanochannel. Moreover, nanochannel with FE_OS can probe targets of sizes beyond the diameter of the channel. I think the results are interesting and have the potential to be published on Nature Communications. Below are the questions and comments I have for the authors to improve the manuscript.

1. There is wide spectrum of nanochannel diameter in both the raw AAO and none@OS cases. What is the effect of such heterogeneity on the performance of the system? In the COMSOL modeling of the nanochannel, idealized geometry and uniform parameters are used. Can the authors justify such assumptions in the model?

2. What are the grafting density and thickness of different polymers, and what are their effects on the ionic transport?

3. The nanochannels in this study are asymmetric by design. Towards a more comprehensive

understanding of the role of FE_OS in ionic transport, it would be instructive to study a symmetric case where both sides have FE_OS. Maybe a simple study can be done using the COMSOL model.

4. The authors have used different FE_OS to functionalize the nanopore, what guidelines can we learn from these case studies regarding the choice of FE_OS for particular applications?

5. I would expect the polymer conformation of the FE_OS, their charge distribution, modification of the electric double layer to impact the ionic transport through the nanopore. More discussion is needed on this topic.

6. The language needs to be improved in general.

Minor comments:

Some references to figures seem to be not accurate. For example, on page 9, "stage nanochannel-system (Fig. 2f, 2g, Fig. S12, S13)." Fig. 2f, 2g should be Fig. 3f, 3g?

Page 11, "Rchannel was almost unchanged with increase of concentration PAA..." Rchannel needs to be defined first.

Page 16, "g, the LoD of ATP" g should be h.

Page 29, "can be solved utilizing appropriate boundary condition." Refer to Eq. 4, 5?

Response to Reviewer #1

(*Q*, the referee's comments; *A*, the authors' response)

Major changes in the revised manuscript are highlighted with blue color.

QI: The manuscript probes the importance of an outer surface of a porous membrane for ionic transport and sensing capabilities of a porous system. I do not believe this work belongs to Nature Communications; importance of the membrane surface and pore walls functionalities on transport in anodic alumina was reported before:

<https://www.nature.com/articles/s41467-017-02447-7>

<https://www.nature.com/articles/s41467-018-06873-z>

AI: We highly appreciated the Reviewer #1's approval on "the importance of an outer surface of a porous membrane for ionic transport and sensing capabilities of a porous system". As the reviewer commented, investigating the contribution from functional elements (FE) at membrane surfaces and pore walls to mass transport is a very important research topic in nanofluidic fields. To systematically study it calls for a series of works rather than one or two papers. In the present work, we explored the importance of membrane functionalities with only functional elements at outer surface (FE_{OS}) and new applications using the nanochannel system on the foundation of two previous works^{1,2} (*Nat. Commun.*, 2018, 9, 40.; *Nat. Commun.*, 2018, 9, 4557.). More importantly, **the importance of the present work is distinct from that of the two previous works.**

In the present work, we demonstrated the role of independent FE_{OS} in the regulating-ion-gating without any contribution from functional elements at inner walls (FE_{IW}) in a nanochannel system (Figure R1a). Different from the present work, our first work demonstrated the synergistic enhancement from FE_{OS} on the regulating-ion-gating of FE_{IW} (*Nat. Commun.*, 2018, 9, 40. <https://www.nature.com/articles/s41467-017-02447-7>) (Figure R1b); the second work demonstrated the anti-interference of FE_{OS} on the regulating-ion-gating of FE_{IW} . (*Nat. Commun.*, 2018, 9, 4557. <https://www.nature.com/articles/s41467-018-06873-z>) (Figure R1c). The difference between the present and the previous works is that the regulation-ion-gating has been mainly achieved by FE_{IW} in the two previous works, while it was achieved by independent FE_{OS} without FE_{IW} in the present work.

(a) Present work

(b) Previous work (*Nat. Commun.*, 2018, 9, 40.)

(c) Previous work (*Nat. Commun.*, 2018, 9, 4557.)

Figure R1. The comparison of present work and two previous works.

The independent usage of FE_{OS} endows the nanochannel system with a series of advantages on regulating ion current in a nanochannel system: (1) **Easy immobilization of FE_{OS} in a nanochannel system.** The outer surface is easier to access for FEs than the inner wall that is a confined space at nanometric scale sometimes even calls for vacuum and electrical driven to realize the transport of functional elements for further functionalities³. (2) **Precise measurement of the physicochemical properties of FE_{OS} .** The FE_{IW} located in the nanoconfined space which is hard to reach for the testing tips or the testing liquids of most current characteristic technologies. The properties of FE_{IW} are hard to be precisely characterized and generally presumed from the properties of the FE_{OS} . In contrast, many techniques can be used in present work to precisely measure the physicochemical properties of FE_{OS} . For example, in the present work, the diameter of nanochannels modified by FE_{OS} , the surface charge density, the distribution of FE_{OS} and the wetting property of OS were precisely measured using atomic force microscopes (AFM), solid surface zeta potential analyzers (SSZPA), the time of flight secondary ion mass spectrometry (ToF-SIMS) and water contact angle measurement respectively, which are hard to be applied for FE_{IW} . (3) **Explicit relationship between the physicochemical properties of FE and ion transport.** In previous works, the relationship between the ion transport and the properties of FE_{IW} is inexplicit due to unmeasurable properties of FE_{IW} in the nanoconfined space. In the present work, we established the explicit relationship between the properties of FE_{OS} and the ion transport by experiments and numerical simulations, which used typical

Poisson and Nernst-Planck (PNP) equations solved with explicit physicochemical parameters of FE_{OS} from actual measurements. Both the qualitative and the quantitative variation of ionic current rectification behaviors from numerical simulations fitted well with the experimental results. (4) **Advantages in the practical applications brought by independent FE_{OS} .** For osmotic energy conversion devices, the output power density of nanochannel systems with independent FE_{OS} was enhanced but the internal resistance doesn't obviously increase; For biochemical sensors, the nanochannel systems with independent FE_{OS} can realized the immobilization of probes with size beyond the diameter of nanochannels and selective detection of targets with size beyond the diameter of nanochannels.

Thanks again for the Reviewer #1's attention to our previous published papers in *Nature Communications* and we also highly appreciate the Reviewer #1's scientific and logical thinking to compare these two works with our present work. In order to justify the importance of the present work, we have modified the relative contents to provide sufficient discussions on the importance of independent FE_{OS} on regulation-ionic-gating in the revised manuscript. The relative contents were listed as below.

“Compared with the confined space in nanochannel, relatively more free spaces of OS endow FE_{OS} with advanced characteristics as easy to immobilize, available for precise characterizations, receptive for foreign substrates.”

Q2: In addition, the manuscript presents a wide spectrum of experimental data with nearly no explanation.

A2: We highly appreciate the reviewer1#’s valuable suggestions. We have added some explanations of experimental data in the revised manuscript as below.

“After attached FE_{OS} , the f_{rec} increased (for PAA@OS and DNA@OS) and the f_{rec} decrease lower than 1 with an opposite polarity (for PEI@OS) (Fig. 3f). This effect was ascribed to the enhancement of negative charge at outer surface from highly negatived PAA rich in hydroxyl in PAA@OS or DNA with phosphodiester skeleton in DNA@OS (Fig. S8), or charge reversal of surface charge for the highly positive PEI@OS rich in amino, leading to the enhancement of ion accumulation and depletion (Fig. S15).”

“The output power density of nanochannel system with independent FE_{OS} was estimated according to the equation $P_L = I^2/R_L$, where I is the current across the circuit and R_L is the external load resistance (Fig. 4a-4d)¹⁵⁻¹⁷.”

“The electricity by reverse electrodialysis is generated under salt gradient using a nanochannel system with only FE_{OS} as separator.”

“It was found that (1) output max power density increased with PAA concentration owing to the enhanced ion selectivity (Fig. 4 e, Fig. S21).”

“The targets were specifically captured by the designed FE_{OS} as probe and tailored the surface charge of the OS locally, which affect the asymmetry of surface potential in between OS and IW and change the ion transport in form of f_{rec} signal. To confirm the sensing mechanism above, we took ATP detection using ssw-DNA as an example. In ssw-DNA structure, one kind of repeating units in ssw-DNA was designed as ATP aptamer, which specifically bonded with ATP and caused the disassembly of

ssw-DNA (Fig. 5d-g, Fig. S29). The variation of surface potential under the disassembly of ssw-DNA triggered by different concentrated ATP was quantitatively characterized through the electrochemical approaches (Fig. 5h, Fig. S30). The selective detection for ATP was also realized based on surface-charge-response sensing mechanism (Fig. 5h and 5i).”

Q3: Figure 2 is incomprehensible.

A3: Thanks for the reviewer 1#'s comment. In order to make the contents of Figure 2 clear, we have replaced the Figure 2 with Figure R2. The part about the definition of FE_{OS} (Figure R2f) was mainly modified.

Figure R2. The characterization of none@OS and the FE_{OS}. **a**, SEM image of none@OS from sectional view. The thickness of nanochannels is 65 μm. **b**, zoom-in version of Au coating side. **c**, The corresponding energy dispersive X-Ray spectroscopy of (b). **d**, **e**, SEM images of the OS coated with Au (d) and without Au (e) of none@OS from top view. **f**, A scheme showing the present nanochannel-system and the FE distribution near the opening of nanochannel-system. The exposed surface

to FE in none@OS includes OS and IW, of which the whole OS and a tiny fraction of IW are attached with FE herein. The FE_{OS} in this work consists of all FE_{OS} and a very small amount of FE_{IW}. **g**, Comparison of the distribution percentage of FE at the IW in the (i) 1st, (ii) 2nd and (iii) 3rd stage. In the 1st stage, the FE occupy the total depth of IW ($\approx 100\%$) through the random indraft of FE (the inset). In the 2nd stage, the distribution percentage of FE at IW decrease down to 5~30% through the Au-S interaction between thiol-modified FE and IW (the inset). In the 3rd stage, the distribution percentage of FE at IW sharply decline near zero. In the (iii) zoom-in version (iv), the distribution percentage of FE decrease with their molar mass, which demonstrates the threshold effects in the 3rd stage.

Q4: It is unclear how the data in Figure 4 was obtained; the Authors for example show values of ion selectivity and maximum power density without any information how this was obtained.

A4: We are genuinely thankful for the Reviewer 1# 's kind remind about the access to the data in Figure 4. Figure 4 mainly discusses the effect from the ion selectivity induced by the FE_{OS} on the osmotic power harvesting of nanochannel systems. The specific description about how to obtain the data has been separately provided in the experimental section and supplementary information of our previous manuscript (Figure S16, S20 and S21 in the revised manuscript). To make readers clearly understand this section, we have rearranged these contents as below and modified the corresponding contents in the revised manuscript and supplementary information.

We set up a test system to investigate nanochannel system's ion selectivity⁴. In this system, KCl solution with salt gradient were placed in the cis (0.5 M) and trans reservoir (0.001 M) which was separated by the nanochannel system. A pair of symmetric Ag/AgCl electrodes were placed in the cis and trans reservoir, respectively (Figure R3). The I-V responses were recorded under a scanning voltage from -0.5 V to 0.5 V.

Figure R3. A scheme of the test system used to investigate nanochannel system's ion selectivity.

According to the previous reports⁴, the ion selectivity can be calculated from osmotic potential (E_{Diff}) using equations:

$$2t_+ - 1 = \frac{E_{\text{Diff}}}{\frac{RT}{zF} \ln \left(\frac{\gamma_{c_H} c_H}{\gamma_{c_L} c_L} \right)}$$

where R , T , z , F , γ , c_H , and c_L represent the gas constant, temperature, charge valence, Faraday constant, activity coefficient of ions, high and low ion concentrations, respectively. For the fixed salt gradient and ambient environment, these parameters are constant. t_+ denotes the transference number for the cation and was regarded as the ion selectivity in the present work. Accordingly, to obtain ion selectivity, we need to

obtain the osmotic potential (E_{Diff}). By recording the I-V response of the nanochannels in the concentration gradient system (Figure R4), the open circuit voltage (E_{Mea} , the potential corresponding to the current of zero) was obtained. According to the equivalent circuit of the testing system in Figure R4, the osmotic potential (E_{Diff}) can be obtained by subtracting the contribution from the electrode-solution interface at salt gradient. The related parameters were listed in Table R1. Then we can calculate the ion selectivity of PAA@OS nanochannel system in Figure 4.

Figure R4. The equivalent circuit of the testing system (a) and the I-V response of the nanochannels in the concentration gradient system (b), where E_{diff} is the osmotic potential and I_{os} is the osmotic current.

Table R1. The related parameters for the nanochannel system modified different PAA concentration.

PAA concentration (mM)	E_{Mea}	E_{Red}	E_{Diff}	t_+
0	102	39	63	0.71

0.01	101	39	62	0.70
0.1	113	39	74	0.75
1	120	39	81	0.77
10	183	39	144	0.97
100	161	39	122	0.90

We further calculated the harvested electric power by transferring it to an external circuit with an electric load resistor (Figure R5). The osmotic power was generated by mixing artificial seawater (0.5 M NaCl) and river water (0.001 M NaCl) via reverse electro dialysis. The output power density of nanochannel system with independent FE_{OS} was calculated according to the equation $P_L = I^2/R_L$, where I is the current across the circuit and R_L is the external load resistance^{5,6}. Then we can calculate the output power density of different nanochannel systems (Figure R6, Figure S20).

Figure R5. A scheme of the test system used to investigate nanochannel system's output power density.

Figure R6. The generated power can be output to external circuit and supply an electronic load.

Q5: It is not clear which advantage the presented system even has over previously reported platforms used for similar sensing mechanisms.

A5: We agree with the Reviewer 1# that the sensing mechanisms using FE_{OS} is similar to that using FE_{IW} in the previous work. However, due to the location at receptive outer surface, the FE_{OS} as probe shows unique advantages over the FE_{IW} as mentioned in the answer to the Q1. Generally, (1) easy to immobilize FE (probes for sensing) (2) the properties of FE (probes for sensing) can be precisely measured. (3) the relationship between the properties of FE (probes for sensing) and ion transport is explicit. (4) When sensing using FE_{OS}, the nanochannel system realized the immobilization of probes with size beyond the diameter of nanochannels for signal amplification and the capture of targets with size beyond the diameter of nanochannels.

As suggested, in order to make a reader fully understanding, we have added related contents in the revised manuscript.

References:

1. Li, X. et al. Role of outer surface probes for regulating ion gating of nanochannels. *Nat. Commun.* **9**, 40 (2018).
2. Gao, P. et al. Distinct functional elements for outer-surface anti-interference and inner-wall ion gating of nanochannels. *Nat. Commun.* **9**, 4557 (2018).
3. Tagliazucchi, M., Rabin, Y. & Szleifer, I. Transport rectification in nanopores with outer membranes modified with surface charges and polyelectrolytes. *ACS Nano* **7**, 9085-9097 (2013).
4. Kim, D., Duan, C., Chen, Y. & Majumdar, A. Power generation from concentration gradient by reverse electrodialysis in ion-selective nanochannels. *Microfluid. Nanofluid.* **9**, 1215-1224 (2010).
5. Gao, J. et al. High-performance ionic diode membrane for salinity gradient power generation. *J. Am. Chem. Soc.* **136**, 12265-12272 (2014).
6. Zhang, Z. et al. Engineering smart nanofluidic systems for artificial ion channels and ion pumps: from single-pore to multichannel membranes. *Adv. Mater.* **32**, 1904351 (2019).

Response to Reviewer #2

(*Q*, the referee's comments; *A*, the authors' response)

Major changes in the revised manuscript are highlighted with blue color.

General Comment:

The manuscript 'Towards Explicit Regulating-Ion-Transport: Nanochannels with only Function Elements at Outer-Surface' presented an interesting manuscript, in which the authors designed nanochannel-system to regulate the ion transport process by using FE_{OS} independently without FE_{IW} . They discuss the effect of outer surface on the ion transport process by using different chemical agents including PAA, PEI, and DNA molecules. The Finite Element Modeling simulation was also conducted to test the correlation of FE_{OS} and I-V result. At the end, this unique nanopore array also was used to achieve osmotic energy conversion and biosensing. All the results in the manuscript show the advantages which the FE_{OS} promote osmotic energy power density and can accommodate targets with size beyond the diameter of channels. The manuscript provides an important advance in ionic channel field, the experimental procedure is thorough with a high number of repeats. For these reasons, I fully support it to be published in Nat. Commun. That said, there were a couple aspects of the work which I felt required some further explanation, clarification or experimental evidence to ensure a reader fully understanding.

Response to General Comment:

We are grateful for the reviewer 2#'s positive evaluation of our work. His/her careful evaluation and detailed comments are very helpful for us to improve the quality of our work. All the revised parts can be found in the revised manuscript with changes marked.

QI: In introduction part. The authors stated that it is necessary and difficult to study the effect of FE_{IW} on ion transport. Whereas, the main purpose of this manuscript is the effect of FE_{OS} rather than FE_{IW} on ion transport. It seems that the authors overemphasize the role of FE_{IW} in ion transport. Thus, it is suggested that the authors polish the introduction part and make the manuscript more readable.

AI: We highly appreciate the reviewer for the valuable suggestions on the content of the introduction. As suggested, we have polished the corresponding contents in introduction part as below.

Page 2, “and the other is inexplicit chemical and physical properties of FE, mainly including FE_{IW} which is subject to that few test techniques with test tips or testing liquids that can sufficiently contacted with FE_{IW} in the confined space at the nanoscale (1st stage in Fig. 1a).” has been revised for “and the other is inexplicit chemical and physical properties of FE located deep inside nanochannel which is subject to that few test techniques with test tips or testing liquids that can sufficiently contacted with FE in the confined space at the nanoscale (1st stage in Fig. 1a).”.

Page 2, “The properties of FE_{IW} , however, are still inexplicit or generally presumed

from the properties of FE_{OS} , which would bring potential errors.” has been revised for “However, till now the properties of all FE are still inexplicit or generally presumed from the properties of measurable FE_{OS} , which would bring potential errors.”.

Page 2, “Even in the 2nd stage, for FE_{IW} in regulating-ion-transport, the “black box” for the properties of FE_{IW} still remained, indicating the 2nd “black box” is related to FE_{IW} , which is hard to tackle.” has been place with ““Even in the 2nd stage, the “black box” for the properties of FE still remained, which is difficult to solve.”.

Page 3, “...but also avoids the deviations or sometimes the errors because the characteristic properties of FE_{IW} cannot be accurately measured in 1st and 2nd stages...” has been revised “...but also avoids the deviations or sometimes the errors because the characteristic properties of FE deep inside nanochannels cannot be accurately measured in 1st and 2nd stages...”.

Page 3, “...these FE_{OS} faithfully realized key capabilities of FE_{IW} on the regulating-ion-transport...” has been revised for “...these FE_{OS} faithfully realized key capabilities of the pervious nanochannel systems on the regulating-ion-transport”.

Q2: The author mentions that PNP equations using the measured parameters of the FE_{OS} , were well fitted with the experiment results in 3rd stage, indicating the explicit regulating-ion-transport accomplished by FE_{OS} without FE_{IW} . What is the advantage of ion-transport-regulation using FE_{OS} only? Can the author list some applications or add more information about ion-transport with only Functions-Elements at Outer-Surface in the introduction?

A2: We highly appreciate the Reviewer #2's suggestions on emphasizing the advantage of FE_{OS}. The independent usage of FE_{OS} endows the nanochannel system with a series of advantages on regulating ion current in a nanochannel system: (1) **Easy immobilization of FE_{OS} in a nanochannel system.** The outer surface is easier to access for FEs than the inner wall that is a confined space at nanometric scale sometimes even calls for vacuum and electrical driven to realize the transport of functional elements for further functionalities¹. (2) **Precise measurement of the physicochemical properties of FE_{OS}.** The FE_{IW} located in the nanoconfined space which is hard to reach for the testing tips or the testing liquids of most current characteristic technologies. The properties of FE_{IW} are hard to be precisely characterized and generally presumed from the properties of the FE_{OS}. In contrast, many techniques can be used in present work to precisely measure the physicochemical properties of FE_{OS}. For example, in the present work, the diameter of nanochannels modified by FE_{OS}, the surface charge density, the distribution of FE_{OS} and the wetting property of OS were precisely measured using atomic force microscopes (AFM), solid surface zeta potential analyzers (SSZPA), the time of flight secondary ion mass spectrometry (ToF-SIMS) and water contact angle measurement respectively, which are hard to be applied for FE_{IW}. (3) **Explicit relationship between the physicochemical properties of FE and ion transport.** In previous works, the relationship between the ion transport and the properties of FE_{IW} are inexplicit due to unmeasurable properties of FE_{IW} in the nanoconfined space. In the present work, we established the explicit relationship between the properties of FE_{OS}

and the ion transport by experiments and numerical simulations, which used typical Poisson and Nernst-Planck (PNP) equations solved with explicit physicochemical parameters of FE_{OS} from actual measurements. Both the qualitative and the quantitative variation of ionic current rectification behaviors from numerical simulations fitted well with the experimental results. (4) **Advantages in the practical applications brought by independent FE_{OS} .** For osmotic energy conversion devices, the output power density of nanochannel systems with independent FE_{OS} was enhanced but the internal resistance doesn't obviously increase; For biochemical sensors, the nanochannel systems with independent FE_{OS} can realized the immobilization of probes with size beyond the diameter of nanochannels and selective detection of targets with size beyond the diameter of nanochannels.

As suggested, we have added more information about the advantage of ion-transport with only functions-elements at outer-surface in the introduction part as below.

“Compared with the confined space in nanochannel, relatively more free-spaces of OS endow FE_{OS} with advanced characteristics, such as easy to immobilize, available for precise characterizations, receptive for foreign substrates and potential application in new scenarios.”

Q3: Figure 1 depicts designed nanochannel-systems attached with FE_{OS} and FE_{IW} at stage1, 2 and stage 3. It is better to label in detail in the figure 1 about black box and the unclear properties or functions. For example, can the authors present the difference between 1st and 2nd. The meaning of ‘Properties’ and ‘Functions’ should be clarified, contributing to the understandable of this figure.

A3: Thanks for the Reviewer #2's suggestions. As suggested, the “black box” and the unclear properties or functions was labeled in detail. The properties represent the physicochemical properties of functional elements and the functions represent the role of functional elements on the regulating ion transport across nanochannels.

Stage	Nanochannel system	Nanochannel	FE _{OS}	FE _{IW}
a 1st				Nanochannel + FE _{OS} + FE _{IW}				
Physicochemical Properties	Partially clear	Clear	Clear	Unclear
Function on Regulating-Ion-Transport	Partially clear	Clear	Unclear	Partially clear
b 2nd				Nanochannel + FE _{OS} + FE _{IW}				
Physicochemical Properties	Partially clear	Clear	Clear	Unclear
Function on Regulating-Ion-Transport	Partially clear	Clear	Partially clear	Clear
c 3rd				
Nanochannel + FE _{OS} (only)				
Physicochemical Properties	Clear	Clear	Clear	
Function on Regulating-Ion-Transport	Clear	Clear	Clear	

Two black shades refer to the two “black boxes” : one is the role of FE_{OS} on regulating-ion-transports and the other is unclear physicochemical properties of FE_{IW}.

Figure R7. The revised Figure 1 as suggested.

Q4: In ‘Explicit role of FEOS on regulating-ion-transport’ section, the author discussed the rectification ratio (frec) measured from I-V tests. How about the ion transport dynamics with FROS only? It is better to illustrate the underlying reason in figure 3b and figure 3f about the correlation of rectification ration and chemical composition.

A4: We highly appreciated the Reviewer 1#’s suggestions. We tried to investigate the ion transport dynamics (1) and establish the relationship between rectification ratio and chemical composition (2), respectively

(1) According to the suggestions, the ion transport dynamics of the nanochannel systems with FE_{OS} only were characterized using the *I-t* test in a KCl electrolyte under a +2 V and -2 V transmembrane voltage, respectively. Four kinds of nanochannel systems have been tested, including the raw nanochannels (none@OS) and the functional nanochannels with FE_{OS} as PAA (PAA@OS), PEI (PEI@OS) and DNA (DNA@OS). The results were shown in Figure R8. For all four sample, under constant voltage driving, the absolute value of ionic current first slightly increases, then decreases and finally keep nearly constant over time.

Figure R8. The ion transport dynamics with FEOS, (a) none@OS, (b) PAA@OS, (c) PEI@OS and (d) DNA@OS, respectively.

The results above arose from the reason as follows. In the nanochannel systems, the ion migration is mainly owing to the electrophoresis driven by the external field. When without any the external bias, ions are randomly distributed throughout the channel, no ionic current exists. If applied a positive bias, cations start to migrate along the direction of electric field and anions migrates opposite direction, leading to an ionic current. With keeping up the external bias, the oppositely charged ions pile up near to the transition region between positive charged region (AAO) and negative charged region (Au), resulting in an ion-induced built-in electric field with the opposite direction to the external electric field, which partially offset the external bias and reduces the ionic current². As more and more ions moving toward and

accumulating near the transition region, the ion-induced electric field continuously increases, leading to a gradual decrease of total current and finally to a stable value when the ion accumulation reaches the equilibrium conditions. For PAA@OS and DNA@OS, the immobilization of PAA and DNA increased the negative charge at the outer surface, leading to more cation accumulation than that of none@OS. As for PEI@OS, the immobilization of PEI induces the transformation of charge polarity at outer surface from negative to positive, leading to the anion accumulation near the outer surface of PEI@OS.

(2) The chemical structure of the functional elements used in this work was shown in Figure R9. Polyacrylic acid (PAA), polyethyleneimine (PEI) and DNA are water-soluble polyelectrolyte. In a neutral pH aqueous solution, PAA is highly negative-charged polyelectrolyte due to the ionization of the pendent carboxyl side chains, PEI is highly positive charged due to the ionization of the pendent amino side chains and DNA is highly negative-charged due to the ionization of the backbone phosphodiester groups. The charged polymer tailors the surface potential of nanochannel and impact on the ionic transport across the nanochannels. (Figure 3e).

Figure R9. The structure of functional elements used in this work.

The underlying reason in figure 3b and figure 3f about the correlation of rectification ration and chemical composition can be ascribed to the difference of charge polarity between the FE_{OS} modified outer surface and the uncovered AAO. The isoelectric point (pI) of AAO membrane is around 8~9, due to the plenty of hydroxyl groups³. In the testing electrolyte (pH = 7.0), the uncover AAO surface inside nanochannel is positively charged. After Au deposition, the outer surface is negatively charged (Fig. 2d), leading to a charge heterojunction, similar to p-n junction. A current rectification behavior was observed, which arose from the ion

enrichment and dissipation in the nanochannel system confirmed by the simulation results (Figure S15). The immobilization of negative-charged PAA or DNA at the OS further enhanced the surface negative charge (Figure 3d), and thus the effect of ion enrichment and dissipation (Figure S15). Therefore, the ion current rectification behavior was enhanced for PAA@OS and DNA@OS (Figure 3f). In contrary, the immobilization of positive-charged PEI at the OS lead to an asymmetric distribution of positive charges along the nanochannel. In the I-V curves, the ion concentration at -2 V was higher than that of 2 V (Figure 3b), which leads to a reverse ion current rectification behavior for PEI@OS comparable to none@OS, PAA@OS and DNA@OS. In a conclusion, the rectification behavior was related to the chemical composition of FE_{OS} that could change the rectification ratio and polarity (Figure 3b), confirmed by the experiments and the numerical simulations (Figure R10).

Figure R10. The f_{rec} calculated by numerical simulations versus of surface charge density at outer surface attached with negative or positive FE.

As suggested, we have added the discussion about the about the correlation of rectification ration and chemical composition as below:

“After attached FE_{OS}, the f_{rec} increased (for PAA@OS and DNA@OS) and the f_{rec} decrease lower than 1 with an opposite polarity (for PEI@OS) (Fig. 3f). This effect was ascribed to the enhancement of negative charge at outer surface from highly negativated PAA rich in hydroxyl in PAA@OS or DNA with phosphodiester skeleton in DNA@OS, or charge reversal of surface charge for the highly positive PEI@OS rich in amino, leading to the enhancement of ion accumulation and depletion.”

Q5: In figure 4a, the author gives a scheme showing the working mechanism of the osmotic energy conversion devices using nanochannel-system but the ‘osmotic energy conversion’ meaning is not obvious. Please add more detailly to make us better understand. With the increasing PAA concentration in precursor, the ion selectivity and the output max power density both increases and has the same tendency. How about the relationship between them?

A5: Thanks for the Reviewer #2’s suggestions. As suggested, we redraw the scheme of Figure 4a and added more details in the scheme, and the revised scheme was shown in Figure R11.

Figure R11. A revised scheme showing the working mechanism of the osmotic energy devices using a nanochannel system containing only FE_{OS}.

In order to make clear for reader, we further added a description of the working mechanism of the osmotic energy conversion devices in the figure caption as followed:

The electricity by reverse electrodialysis is generated under salt gradient using a nanochannel system with only FE_{OS} as separator.

We can establish the relationship between the output max power density and the ion selectivity of the membrane according to the equation of the theoretical output max power⁴ (P_{max}) as

$$P_{max} = \frac{1}{4} \frac{E_{Diff}^2}{R_{channel}} \quad (2)$$

Where E_{Diff} is the pure osmotic potential, contributed by the cation-selective nanochannel systems, and $R_{channel}$ is the internal resistance of nanochannels. Meanwhile, the diffusion potential (E_{Diff}) satisfies the following equation:

$$E_{Diff} = (2t_+ - 1) \frac{RT}{zF} \ln \left(\frac{\gamma_{c_H} c_H}{\gamma_{c_L} c_L} \right) \quad (3)$$

where R , T , z , F , γ , c_H , and c_L represent the gas constant, temperature, charge valence, Faraday constant, activity coefficient of ions, high and low ion concentrations, respectively. For fixed salt gradient and ambient environment, these parameters are constant. t_+ denotes the transference number for the cation and is defined as ion selectivity in our system. Hence, according to the equation (2) and (3), the ion selectivity and the theoretical output max power density will increase with the ion selectivity, which is in accordance with the experiment results in the present work.

Q6: It is better to add more words about ‘Impact from FEOS on biosensors’ section to show that the author has realized a nearly ‘universal’ biosensor approach: one is the DNA amplifications as the probes and the other is the MCF7 cells as the targets. The mechanism of this biosensor should also be presented, why do the presence of target would trigger the changes in ion rectification ration? Related FEM simulation is needed here to support the experimental result and speculation.

A6: Thanks for the Reviewer #2’s suggestions. As suggested, we added the sensing mechanism of this biosensors and their advantage as a nearly ‘universal’ biosensor approach in the revised manuscript as below:

“The targets were specifically captured by the designed FE_{OS} as probe and tailored the surface charge of the OS locally, which affect the asymmetry of surface potential in between OS and IW and change the ion transport in form of f_{rec} signal. To confirm the sensing mechanism above, we took ATP detection using ssw-DNA as an example. In ssw-DNA structure, one kind of repeating units in ssw-DNA was designed as ATP aptamer, which specifically bonded with ATP and caused the disassembly of ssw-DNA (Fig. 5d-g, Fig. S29). The variation of surface potential under the disassembly of ssw-DNA triggered by different concentrated ATP was quantitatively characterized through the electrochemical approaches (Fig. 5h, Fig. S30). The selective detection for ATP was also realized based on surface-charge-response sensing mechanism (Fig. 5h and 5i). In the 1st and 2nd stage, because probes (as FE_{IW}) were immobilized at the IW, a confined space usually with diameter < 100 nm, the targets with a size beyond the diameter of nanochannels can’t sufficiently contact with

probe and efficiently recognized. In the 3rd stage, the OS possess the receptive characteristic for probes or targets with the size beyond the diameter of nanochannels. The recognition between probes and targets took place at the OS, which is relatively more free-spaces compared with IW in nanochannels.”

We also took the ATP detection using ssw-DNA as probe and did the related FEM simulations to verify the experimental results and our speculations on sensing mechanism. When using the measured parameter from experiments, the result of numerical simulation is in accordance with the experimental results (Figure R12).

Figure R12. The ATP detection using ssw-DNA as probe. (a) Formation of “supersandwich” DNA structure (ssw-DNA) with long concatamers through the successive hybridization of alternating DNA unit. (b) LoD of ATP using ssw-DNA as probe based on electrochemical method. (c) The experimental results and corresponding theoretical results.

Q7: In figure 5c, this nanochannel-system can detect MCF7 cells. Could the author explain the reason why can this device accommodate targets with size beyond the diameter of nanochannels? Why does the percentage of MCF7 cells is larger than Hela cell and PC3 cell? The legend of figure 5c could be written more detailly.

Q7: We reckon that there are three primary causes of the accommodation of the device with size beyond the diameter of nanochannels. First, compared with the confined space inside nanochannel (with diameter < 100 nm), the OS offer a relative free space to accommodate targets. Second, diverse probes are easy to be immobilized at the OS, which can be used for the specific capture of diverse targets. Third, as mentioned above, the capture of electrical charged targets at the OS or the variation of OS potential triggered by targets can impact on the ionic rectification and produce f_{rec} signal output.

The MCF-7 cells detection was realized based on the specific interaction between the MUC1 mucin on the surface of MCF-7 cells and its aptamer (S2.2, GCAGTTGATCCTTTGGATACCCTGG) immobilized at the OS. MUC1 mucin is a large transmembrane glycoprotein, whose expression increases at least 10-fold at the surface of MCF-7 cells in primary and metastatic breast cancers, which makes it an ideal target molecule for the detection of MCF-7 cells⁵. S2.2 is a 25-base oligonucleotide that binds to MUC1 protein with high affinity and specificity. Compared with MCF-7 cells, Hela and PC3 cells did not interact with the S2.2 aptamer, thus did not change the surface charge at outer surface. Therefore, the percentage of MCF7 cells is larger than Hela cell and PC3 cell.

As suggested, we have revised the legend of the Figure 5c and added more description.

“The selective detection of multi-scale targets using FE_{OS} was realized based on the change of f_{rec} signal output induced by the surface charge at outer surface.”

Q8: The physical model and details (e.g. Geometrical Model, Boundary setting, and Mesh Parameters) of FEM simulation should be presented in supporting information. Furthermore, the related explanation for PNP equation also should be listed.

A8: We appreciate the Reviewer 3# 's careful survey. As suggested, we have added the physical model and details of numerical simulation in supporting information as below.

“The 2D axisymmetric geometry of a nanochannel for the simulation was shown in Figure S14. The total length of nanochannel, FE and Au were scaled down at the same proportion based on the measured parameters due to the limitation of simulations. Two reservoirs were set to keep the system stable. In the simulation, the pore diameter was set to be 25 nm and the pore diameter for none@OS, PAA@OS, PEI@OS and DNA@OS side was used according to the measured parameters. An adaptive mesh refinement was used to optimize the mesh size geometry. The boundary conditions for eqs 1 to 3 are summarized in Table S2, using the numbers 1-14 to designate the surfaces defining the model system.”

Table S2. Boundary Conditions for the Numerical Solution

Surface	Nernst-Planck Equation	Poisson Equation
---------	------------------------	------------------

	eq 1	eq 2
1	$c(K^+) = c(Cl^-) = c_0$	V_b V
2	$c(K^+) = c(Cl^-) = c_0$	0 V
3, 4, 5, 6	$\vec{n} \cdot \vec{j}_i = 0$	Zero charge
7, 8, 9, 10, 11, 12, 13, 14	$\vec{n} \cdot \vec{j}_i = 0$	$\vec{n} \cdot \nabla \varphi = -\frac{\sigma_s}{\varepsilon}$

Furthermore, as suggested, the related explanation for PNP equation was also supplemented in the revised manuscript as below.

“Equation (1) is the Nernst-Planck equation that describes the transport properties of a charge nanochannel and equation (2) is the Poisson equation that describes the relationship between the electric potential and ion concentration in the nanochannels. The model is generally simplified using steady-state conditions (equation (3)).”

A9: As the authors mentioned in the manuscript, the behavior of ion transport in a confined space is different from that in bulk solution. For example, the asymmetric geometry of confined space and asymmetric distribution of charge in confined space contribute to the phenomenon of ion current rectification. Besides, the electrochemically confined effect within the nanochannel or nanopore allows the single molecule sensing with high spatial and temporal resolution. Thus, I wonder can the author make comments on the electrochemistry in a confined space? And demonstrate the effect of confined effect on the ion transport process in detail.

A9: Thanks for the Reviewer #2's suggestions. As suggested, we examined the literatures about the electrochemistry in a confined space^{6,7}. We have supplemented the comments in the revised manuscript as below.

“A nanochannel system combining with electrochemistry in a confined space is now a crucial promising field^{6,7}, which is utilized to dynamically monitor the single molecule⁸, understand the chemical reaction⁹, characterize the single particle¹⁰, and probe single living cell¹¹, etc. ”

The references used above was numbered ref 45-50 in the revised manuscript.

Q10: There are also some format issues the authors should notice and check carefully:

- 1) The volume of journal should be bold in reference 43 and reference 44.
- 2) The journal name should be abbreviated as ‘Angew. Chem. Int. Ed.’ in reference 26.

A10: Thanks a lot for the Reviewer 3# 's careful reading. We have revised the format issues of references throughout the manuscript.

26. Liu, N. et al. Two-way nanopore sensing of sequence-specific oligonucleotides and small-molecule targets in complex matrices using integrated DNA supersandwich structures. *Angew. Chem. Int. Ed.* **52**, 2007-2011 (2013).

43. Wang, Y., Yang, Q., Zhao, M., Wu, J. & Su, B. Silica-Nanochannel-Based interferometric sensor for selective detection of polar and aromatic volatile organic compounds. *Anal. Chem.* **90**, 10780-10785 (2018).

44. Sun, Y. et al. A highly selective and recyclable NO-responsive nanochannel based

on a spiroring opening-closing reaction strategy. *Nat. Commun.* **10** (2019).

References:

1. Tagliazucchi, M., Rabin, Y. & Szleifer, I. Transport rectification in nanopores with outer membranes modified with surface charges and polyelectrolytes. *ACS Nano* **7**, 9085-9097 (2013).
2. Li, D., Wu, H., Cheng, H., Wang, G., Huang, Y. & Duan, X. Electronic and ionic transport dynamics in organolead halide perovskites. *ACS Nano* **10**, 6933 (2016).
3. Gao, J. et al. High-performance ionic diode membrane for salinity gradient power generation. *J. Am. Chem. Soc.* **136**, 12265-12272 (2014).
4. Kim, D., Duan, C., Chen, Y. & Majumdar, A. Power generation from concentration gradient by reverse electro dialysis in ion-selective nanochannels. *Microfluid. Nanofluid.* **9**, 1215-1224 (2010).
5. Wu, P., Gao, Y., Zhang, H. & Cai, C. Aptamer-guided silver-gold bimetallic nanostructures with highly active surface-enhanced Raman scattering for specific detection and near-infrared photothermal therapy of human breast cancer cells. *Anal. Chem.* **84**, 7692-7699 (2012).
6. Lu, S., Peng, Y., Ying, Y. & Long, Y. Electrochemical sensing at a confined space. *Anal. Chem.* **92**, 5621-5644 (2020).
7. Ying, Y., Gao, R., Hu, Y. & Long, Y. Electrochemical confinement effects for innovating new nanopore sensing mechanisms. *Small Methods* **2**, 1700390 (2018).
8. Qing, Y., Tamagaki-Asahina, H., Ionescu, S., Liu, M. & Bayley, H. Catalytic

- site-selective substrate processing within a tubular nanoreactor. *Nat. Nanotechnol.* **14**, 1135-1142 (2019).
9. Cao, J., Jia, W., Zhang, J. *et al.* Giant single molecule chemistry events observed from a tetrachloroaurate(III) embedded *Mycobacterium smegmatis* porin A nanopore. *Nat. Commun.* **10**, 5668 (2019).
10. Gao, R., Ying, Y., Li, Y., Hu, Y., Yu, R., Lin, Y. & Long, Y. A 30 nm nanopore electrode: facile fabrication and direct insights into the intrinsic feature of single nanoparticle collisions. *Angew. Chem. Int. Ed.* **57**, 1011-1015 (2018).
11. Nadappuram, B., Cadinu, P., Barik, A., *et al.* Nanoscale tweezers for single-cell biopsies. *Nat. Nanotechnol.* **14**, 80-88 (2019).

Response to Reviewer #3

General Comment:

In this paper, the authors investigated the impact of functional elements at the outer surface (FE_{OS}) of nanopores on the regulation of ionic transport. By depositing Au on one side of the system, shrinkage of the pore exit is achieved, which limits the functional polymers to anchor to the inner surface of the nanochannel. As a result, the authors were able to focus on the role of FE_{OS} in the function of the nanochannel. The experiment showed that the FE_{OS} enhance the power density and ionic selectivity of the nanochannel. Moreover, nanochannel with FE_{OS} can probe targets of sizes beyond the diameter of the channel. I think the results are interesting and have the potential to be published on Nature Communications. Below are the questions and comments I have for the authors to improve the manuscript.

Response to General Comment: We highly appreciate the reviewer's positive comments on our work. His/her suggestions are really helpful for improving the quality of our present work. We have done our best to comply with them in the revised manuscript. The following are our responses to their comments one by one. All revised parts can be found in the revised manuscript with changes marked.

Q1: There is wide spectrum of nanochannel diameter in both the raw AAO and none@OS cases. What is the effect of such heterogeneity on the performance of the system? In the COMSOL modeling of the nanochannel, idealized geometry and uniform parameters are used. Can the authors justify such assumptions in the model?

AI: Thanks for the Reviewer #3's suggestions. As suggested, we have supplemented the numerical simulations about the effect from the wide spectrum of nanochannel diameter on the performance of the present system. To investigate the effect, we first set up multi-channel model (Figure R13) and then change the heterogeneous degree of nanochannel diameter (Table R3, Figure R14). In the first step, with increasing channel number in the model, the ionic current linearly increased, while the rectification ratio (f_{rec}) of ionic current slightly decreased. It demonstrated that our multi-channel model can run. In the second step, we changed the heterogeneous degree of nanochannel diameter according to the range from 20 to 30 nm for AAO (25 ± 5 nm) and from 8 to 14 nm for Au (11 ± 3 nm). There are also two kinds of diameter distribution as (1) uniform distribution and (2) bipolar distribution. Taking Au (11 ± 3 nm) for example, the uniform distribution means 8 nm for "channel 1", 9 nm for "channel 2", 10 nm for "channel 3", 11 nm for "channel 4", 12 nm for "channel" 5, 13 nm for "channel 6", while, the bipolar distribution means, 8 nm for "channel 1-3" and 15 nm for "channel 4-6". According to the simulation results, the rectification behavior of nanochannels was well kept under the effect of such heterogeneity. There is a float of the rectification ratio. Compared with the experimental result in the present work, the float of the rectification ratio from

simulation is approximate with the error bar from experimental results. It indicated that the wide spectrum of nanochannel diameter didn't change the rectification nature of this system, while, bring the float of rectification ratio.

Figure R13. The theoretical results using multi-channel models.

Table R3. The theoretical results using a wide spectrum of nanochannel diameter.

Model	Au	AAO	I_{+2V}	I_{-2V}	f_{rec}
	diameter (nm)	diameter (nm)			
(4)	11	25	0.06471	-0.02421	2.67
	11	25			
	11	25			
	11	25			
	11	25			

(5)	11	30	0.06498	-0.02480	2.62
	11	30			
	11	30			
	11	25			
	11	25			
	11	25			

(6)	8	25	0.06614	-0.02423	2.73
	8	25			
	8	25			
	11	25			
	11	25			
	11	25			

(7)	8	25	0.06412	-0.02466	2.60
	8	25			
	8	25			
	14	25			
	14	25			
	14	25			

(8)	11	30	0.06498	-0.02470	2.63
	11	30			
	11	30			
	11	20			

11 20

11 20

(9)	8	30	0.06395	-0.02459	2.60
	8	30			
	8	30			
	14	20			
	14	20			
	14	20			

8 20

9 22

(10) 10 24 0.06405 -0.02473 2.59

11 26

12 28

14 30

Figure R14. The theoretical results using multi-channel models with a wide spectrum of nanochannel diameter in both the raw AAO and none@OS cases.

As the comment that “In the COMSOL modeling of the nanochannel, idealized geometry and uniform parameters are used. Can the authors justify such assumptions in the model?”, we have justified such assumptions as below.

In this work, the AAO channel was cylindrical shape demonstrated by the SEM. While, the geometry cannot be observed. In order to confirm the rationality of this assumption, we have performed the numerical simulations using an asymmetric Au nanochannel and symmetric AAO nanochannel. The theoretical results demonstrated that the asymmetric geometry of Au nanochannel did not significantly affect the results in the represent model due to that the Au layer is thin, but should come into notice when Au layer is thick (Figure R15).”

Figure R15. The theoretical results using cylindrical and conical Au nanochannel in the models.

Q2: What are the grafting density and thickness of different polymers, and what are their effects on the ionic transport?

A2: We highly appreciated the idea proposed by the Reviewer 3#. Both the grafting density and thickness of functional polymers has been certified with great impact on the ionic transport in 1st and 2nd stage. Until now, these effects have not been investigated in the case of the polymer located at the outer surface. According to the suggestions of the Reviewer 3#, we did a series of experiments including (1) quantitative investigation on the effect from DNA grafting density and thickness and (2) qualitative investigation on the effect from PEI and PAA grafting density and thickness on the ionic transport.

(1) For the effect from polymer grafting density, we firstly change the grafting density of DNA on the OS by adding different concentrated DNA solution. The grafting density of DNA was measured by a classical electrochemical method (chronocoulometry)^{1, 2}. In short, Hexaammineruthenium(III) chloride (RuHex) molecule was used for the accurate characterization of DNA assembly amount and density. When RuHex fully adsorbed in DNA backbone, the amount of RuHex and the corresponding base number of DNA complexes can be calculated by the integral area of the peak in chronocoulometry technique. Because of the nearly all the synthetic DNA with 35 base number, the grafting density of DNA with different concentration at outer surface was calculated as 1.86×10^{12} molecule/cm², 3.23×10^{12} molecule/cm² and 3.52×10^{12} molecules/cm², respectively (as shown in Fig. R16a). Then, we investigated the impact from the DNA grafting density at OS on the ionic transport. We found that the f_{rec} increased with the increase of grafting density of DNA molecules (Figure R16b).

Figure R16. The electrochemical characterization of different DNA grafting density at outer surface and the effect of DNA with different grafting density on the ion current rectification ratio.

We secondly used the same method to change the grafting density of PAA and PEI through changing the concentration of polymer in precursor. The grafting density of PAA and PEI increased with their concentration in precursor and f_{rec} increased as the increase of grafting density of PAA and PEI (Figure R17). In a conclusion, we found that the f_{rec} was positively related to the grafting density of FE_{OS} .

Figure R17. The effect of different functional elements (PAA or PEI) on the ion current rectification ratio.

(2) For the effect from polymer thickness, we firstly changed the thickness of DNA through the disassembly of supersandwich structured DNA. Taking ATP detection using ssw-DNA as an example, one kinds of repeating units in ssw-DNA was designed as ATP aptamer, which specifically bonded with ATP and caused the disassembly of ssw-DNA (Figure R18a). The variation of surface potential under the disassembly of ssw-DNA triggered by different concentrated ATP was quantitatively characterized through the electrochemical approaches (Figure R18b). With the disassembly of sswDNA, and the reduction of thickness of DNA functional layer, the f_{rec} decreased.

Figure R18. The effect from the thickness of DNA functional elements on the ion current rectification ratio taking the disassembly of ssw-DNA as example.

As suggested, we have also studied the effect of the thickness of PAA or PEI on the f_{rec} of nanochannel systems by continuous deposition PAA or PEI on the outer surface. The thickness of the PAA and PEI functional layer was measured using Ellipsometer³ (Figure R19). We found that the thickness of the polymer didn't change obviously by overlay the same polymer (Fig. R19a), which may be attributed to the fact that the electrostatic repulsion obstacle the assembly between with negative-charged PAA or positive-charged PEI. We found the f_{rec} kept almost constant after overlay. The two experiments indicated that the f_{rec} increase with the thickness of FE_{OS} layer.

Figure R19. The impact of the thickness of PAA and PEI on the ion current rectification ratio.

Q3: The nanochannels in this study are asymmetric by design. Towards a more comprehensive understanding of the role of FE_{OS} in ionic transport, it would be instructive to study a symmetric case where both sides have FE_{OS}. Maybe a simple study can be done using the COMSOL model.

A3: We highly appreciate the reviewer3#'s valuable suggestions. To study a symmetric case where both sides have FE_{OS} is a very interesting research topic. As suggested, we preliminarily explored the effect of FE immobilized on both outer surface of nanochannel systems by numerical simulations. A symmetric model where both sides have FE_{OS} was designed (Figure R20). As a result, the ion current was

linear and the ion current rectification was not observed for PAA@PAA or PEI@PEI, two symmetric cases. The ion current was not affected by the immobilization of PAA at both sides of nanochannel, while the ion current decreased for the PEI@PEI (Figure R20b and d). For the PAA@PEI, the asymmetric ion transport was observed (Figure R20f), an ion current rectification behavior, owing to the asymmetric charge distribution along the nanochannel.

Figure R20. The theoretical results using a symmetric case where both outer surfaces have FE_{OS}.

Q4: The authors have used different FE_OS to functionalize the nanopore, what guidelines can we learn from these case studies regarding the choice of FE_OS for particular applications?

A4: We highly appreciate the reviewer's valuable suggestions.

Generally, the guidelines for the choice of functional elements in this work was summarized as (1) the molecule size is beyond that the pore diameter so that the molecule does not enter into the nanochannel owing the blockage effect, (2) the molecule should bring plenty of charges in the molecule chain so that the ion current rectification can be obtained owing to the asymmetric charge distribution along the nanochannels and (3) the bonding mode between functional elements and outer surface should be careful designed so that the nanochannel system keeps stable in the practical application.

We summarized the guidelines to choose the functional elements according to the three application as ionic rectification, osmotic energy harvesting and biochemical sensing in the present work. Firstly, for the adjustment of ion transport, the guideline we need to consider is the charge polarity, hydrophilic or hydrophobic property and the molecular weight of functional elements. The charge polarity at outer surface plays a key role in the regulation of ion current rectification in nanochannel systems and the rectification ration was close related to the surface charge density. The immobilization of negative FE would enhance the rectification and positive FE would induce the reversal of rectification polarity. Meanwhile, the charge quantity of FE was also important on the rectification ratio.

Secondly, for the application of osmotic energy harvesting, the ion selectivity and the internal resistance are two key factors affecting the energy conversion properties of nanochannel systems. The immobilization of functional elements at outer surface should enhance the energy conversion efficiency. The energy conversion efficiency is closely related to the ion selectivity, which was dependent on the surface properties, surface charge density, etc. Therefore, high surface charge at the outer surface induced by high-charged FE is helpful to enhance the ion selectivity. Meanwhile, the molecule weight of FE needs to ensure that the FE does not enter into the nanochannel and lead to the increase of internal resistance of nanochannel system.

Thirdly, for the biochemical sensing, the aptamer was used for the selective detection of targets. The aptamer using as FE should have high-selective recognition ability for the targets. The combination between FE probes and targets at outer surface should change surface properties of outer surface so that the detection signal can be obtained. If the detection signal is not enough large to be detect, in some cases, the amplified reaction for the FE for the recognition of targets can be used for the enhance detection signal.

Q5: I would expect the polymer conformation of the FE_OS, their charge distribution, modification of the electric double layer to impact the ionic transport through the nanopore. More discussion is needed on this topic.

A5: We highly appreciate the reviewer3#'s valuable suggestions. We designed and did new experiments to investigate the impact from the conformation, charge distribution and electric double layer on the ionic transport.

Firstly, we decorated poly(N-isopropylacrylamide) (PNIPAAm), a thermally responsive functional molecule at the OS, After heating, the conformation of NIAAPM change⁴ (Sun, T., et al., *Angew. Chem., Int. Ed.*, 2004, 43, 357.). Here, we test the contact angle of water at the OS functionalized by NIPAAM at 25° to 50°.

The water contact increased from $42.3^\circ \pm 1.7^\circ$ to $72.6^\circ \pm 2.4^\circ$ which further confirms the conformation variation of NIPAAM as FEOS. However, the current did not change obviously which may be attributed to the low amount of charge of NIPAAM and tiny variation of surface charge after the conformation variation of NIAAM. (Figure R21)

Figure R21. The effect of the conformation of FE_{OS} on the ion transport.

Secondly, the charge distribution of FE_{OS} show great impact on the ion transport. The effect can be confirmed by the experiment results in Figure R18. The current rectification ratio decreased with the diminution of surface distribution at the outer surface by ATP-triggered reduction of ssw-DNA chain.

Finally, according to the equation ($\kappa = \frac{1}{\lambda_d} = \sqrt{\frac{e^2 \sum n_i^0 z_i^2}{\epsilon_0 \epsilon_r k_B T}}$), the double electric layer was mainly affected by the ion concentration of electrolyte. We changed the double electric layer by the changing the concentration of K⁺ in electrolyte. We studied the effect of double electric layer on ion transport by recording the ion current versus the ion concentration of electrolyte. The results were shown in Figure R22. The ion current was closely related to the ion strength for both none@OS and PAA@OS (Figure R22c), which indicated that the electric double layer impact on the ionic transport of the nanochannel system with independent FE_{OS}.

Figure R22. The effect of the ion strength on the ion transport across nanochannels.

Q6: The language needs to be improved in general.

A6: Thanks a lot for the Reviewer 3# 's careful reading and the grammar issues have been corrected as Reviewer 3# 's suggestions. Not only this, we do more corrections to improve the language throughout the manuscript. Grammar mistakes have been listed in the response letter. The articles, tense, singular & plural have been checked throughout the manuscript. All corrections are highlighted in blue.

Page 2, "...that can sufficiently contacted with..." has been revised for "...that can sufficiently contact with...".

Page 3, "...bring two new features in..." has been revised for "...brought two new features in...".

Page 3, "OS referring to the outermost..." has been revised for "OS refers to the outermost...".

Page 4, "...meaning that the stacking FE initially at the opening..." has been revised for "...meaning that the stacking FE initial at the opening...".

Page 8, the title of the third section has been replaced with "Explicit role of FEOS in regulating-ion-transport".

Page 8, "...the f_{rec} decrease lower than..." has been revised for "...the f_{rec} decreased lower than...".

Minor comments:

Some references to figures seem to be not accurate. For example, on page 9, "stage nanochannel-system (Fig. 2f, 2g, Fig. S12, S13)." Fig. 2f, 2g should be Fig. 3f, 3g?

Page 11, "Rchannel was almost unchanged with increase of concentration PAA..."
Rchannel needs to be defined first.

Page 16, "g, the LoD of ATP" g should be h.

Page 29, "can be solved utilizing appropriate boundary condition." Refer to Eq. 4, 5?

Response to minor comments: Thanks a lot for the Reviewer 3# 's careful reading and the related errors have been corrected as Reviewer 3# 's suggestions.

On page 9, the references have been modified to be Fig. 3f, 3g.

Page 11, $R_{channel}$ has been defined as the internal resistance of the nanochannel system.

Page 14, "The numerical simulation of two samples above, PAA4PEI3@OS and

PAA4PEI4@OS...” has been revised for “The numerical simulation of two samples above, PAA4PEI3@OS and PAA4PEI4@OS...”.

Page 16, In the caption of Fig. 5, the error has been modified.

Page 29, The coupled equation (1)-(3) can be solved utilizing appropriate boundary conditions (eq4, eq 5, Table S4).

The format of some references has also been revised.

26. Liu, N. et al. Two-way nanopore sensing of sequence-specific oligonucleotides and small-molecule targets in complex matrices using integrated DNA supersandwich structures. *Angew. Chem. Int. Ed.* **52**, 2007-2011 (2013).

43. Wang, Y., Yang, Q., Zhao, M., Wu, J. & Su, B. Silica-Nanochannel-Based interferometric sensor for selective detection of polar and aromatic volatile organic compounds. *Anal. Chem.* **90**, 10780-10785 (2018).

44. Sun, Y. et al. A highly selective and recyclable NO-responsive nanochannel based on a spiroring opening-closing reaction strategy. *Nat. Commun.* **10** (2019).

References:

1. Steel, A. B., Herne, T. M. & Tarlov, M. J. Electrochemical quantitation of DNA immobilized on gold. *Anal. Chem.* **70**, 4670-4677 (1998).
2. Li, X. et al. Role of outer surface probes for regulating ion gating of nanochannels. *Nat. Commun.* **9**, 40 (2018).
3. Habicht, J., Schmidt, M., Ruhe, J. & Johannsmann, D. Swelling of thick polymer brushes investigated with ellipsometry. *Langmuir* **15**, 2460-2465 (1999).
4. Sun, T., Wang, G., Feng, L., Liu, B., Ma, Y., Jiang, L. & Zhu, D. Reversible

switching between superhydrophilicity and superhydrophobicity. *Angew. Chem., Int. Ed.*, **43**, 357-360 (2004).

REVIEWERS' COMMENTS

Reviewer #1 (Remarks to the Author):

The Authors addressed all my concerns, and I am happy to recommend the manuscript for publication.

Reviewer #2 (Remarks to the Author):

The authors give very responsive response to my comments from that I learn a lot. Thank you ! Thus, I totally recommend the publiation in Nat. Commun.

Reviewer #3 (Remarks to the Author):

I found the revised manuscript improved and ready for publication.

Reviewer #1 (Remarks to the Author):

The Authors addressed all my concerns, and I am happy to recommend the manuscript for publication.

We are thankful to Reviewer #1's laborious effort on our manuscript.

Reviewer #2 (Remarks to the Author):

The authors give very responsive response to my comments from that I learn a lot.

Thank you! Thus, I totally recommend the publication in Nat. Commun.

We are thankful to Reviewer #2's laborious effort on our manuscript.

Reviewer #3 (Remarks to the Author):

I found the revised manuscript improved and ready for publication.

We are thankful to Reviewer #3's laborious effort on our manuscript.